# Targeting epigenetic regulators to overcome drug resistance in the emerging human fungal pathogen *Candida auris*

Yuping Zhang[1,2,7], Lingbing Zeng[1,7], Xinhua Huang[2], Yuanyuan Wang [2], Guangsheng Chen[2], Munika Moses[2], Yun Zou[2], Sichu Xiong[2], Wenwen Xue[3], Yanmei Dong[4], Yueru Tian[5], Ming Guan[5], Lingfei Hu[6], Zhe Yin[6], Dongsheng Zhou [6] ✉, Xiaotian Huang[1] ✉ & Changbin Chen [2,3] ✉

The rise of drug-resistant fungal species, such as *Candida auris*, poses a serious threat to global health, with mortality rates exceeding 40% and resistance rates surpassing 90%. The limited arsenal of effective antifungal agents underscores the urgent need for novel strategies. Here, we systematically evaluate the role of histone H3 post-translational modifications in *C. auris* drug resistance, focusing on acetylation mediated by Gcn5 and Rtt109, and methylation mediated by Set1, Set2, and Dot1. Mutants deficient in these enzymes exhibit varying degrees of antifungal drug sensitivity. Notably, we discover that *GCN5* depletion and the subsequent loss of histone H3 acetylation downregulates key genes involved in ergosterol biosynthesis and drug efflux, resulting in increased susceptibility to azoles and polyenes. Additionally, Gcn5 regulates cell wall integrity and echinocandin resistance through the calcineurin signaling pathway and transcription factor Cas5. In infection models using *Galleria mellonella* and immunocompromised mice, *GCN5* deletion significantly reduces the virulence of *C. auris*. Furthermore, the Gcn5 inhibitor CPTH$_2$ synergizes with caspofungin in vitro and in vivo without notable toxicity. These findings highlight the critical role of Gcn5 in the resistance and pathogenicity of *C. auris*, positioning it as a promising therapeutic target for combating invasive fungal infections.

Infections caused by pathogenic fungi are often associated with various diseases in humans, ranging from mucocutaneous to systemic life-threatening infections. These infections have a profound impact on global health, affecting over 1 billion individuals worldwide[1].

Invasive fungal infections, particularly those caused by common pathogens such as *Candida spp.*, *Cryptococcus spp.*, *Aspergillus spp.*, and *Pneumocystis spp.*, are recognized as hidden killers, with mortality rates often exceeding 50%. These infections lead to ~2.5 million deaths

[1]School of Basic Medical Sciences, and the First Affiliated Hospital, Jiangxi Medical College, Nanchang University, Nanchang, Jiangxi, China. [2]Joint Laboratory for Biomedical Research and Pharmaceutical Innovation, Unit of Pathogenic Fungal Infection & Host Immunity, Key Laboratory of Molecular Virology and Immunology, Shanghai Institute of Immunity and Infection, Chinese Academy of Sciences, Shanghai, China. [3]Nanjing Advanced Academy of Life and Health, Nanjing, China. [4]Department of Gastroenterology and Hepatology, Characteristic Medical Center of the Chinese People's Armed Police Force, Tianjin Key Laboratory of Hepatopancreatic Fiberosis and Molecular Diagnosis & Treatment, Tianjin, China. [5]Department of Laboratory Medicine, Huashan Hospital North, Shanghai Medical College, Fudan University, Shanghai, China. [6]State Key Laboratory of Pathogen and Biosecurity, Academy of Military Medical Sciences, Beijing, China. [7]These authors contributed equally: Yuping Zhang, Lingbing Zeng. ✉e-mail: zhouds@bmi.ac.cn; dongshengzhou1977@gmail.com; xthuang@ncu.edu.cn; cbchen@ips.ac.cn

annually, posing a substantial threat to global public health[1,2]. Thus, there is an urgent need to identify targets for safe and effective countermeasures, such as drugs, vaccines, and diagnostics, that may help reduce in-hospital mortality rates associated with invasive fungal infections.

The emerging "super fungus" *Candida auris* was first isolated in 2009 from an ear canal of a Japanese patient and has since spread across >40 countries on six continents[3]. A direct consequence of invasive *C. auris* infection is the risk of cross-infection among patients and hospital outbreaks due to its remarkable ability to adhere to both biological and non-biological surfaces, its long-term survival under extreme conditions, and its resistance to conventional disinfection methods[4,5]. For instance, since the onset of the COVID-19 pandemic, an increased number of *C. auris* infections has been reported in COVID-19 intensive care units (ICUs) in countries such as the United States, Germany, India, and Mexico[6–9]. Although statistics may vary across reports, mortality rates attributable to invasive *C. auris* infections can reach as high as 70%[10], posing a significant threat and a formidable challenge to the medical field. Notably, *C. auris*, along with *Candida albicans*, *Cryptococcus neoformans*, and *Aspergillus fumigatus*, has been classified in the critical priority group of fungal pathogens in the World Health Organization's (WHO) recently released fungal priority pathogen list[11].

The recent emergence of fungi with inherent or acquired resistance to one or more antifungal drugs has become a serious concern. Currently, only three major classes of antifungal drugs, including azoles (e.g., fluconazole), polyenes (e.g., amphotericin B), and echinocandins (e.g., caspofungin), are available in clinical practice for the treatment of invasive fungal infections[12,13]. Recent studies on the multidrug resistance of *C. auris* have revealed that over 90% of clinical isolates are resistant to fluconazole, 30-41% are resistant to two classes of antifungal drugs, and approximately 4% are resistant to three or more[5,14,15]. Furthermore, *C. auris* can rapidly acquire resistance under the selective pressure of antifungal treatments or other specific conditions[16–19]. For instance, isolates from patients receiving caspofungin treatment have been shown to develop resistance to echinocandins[19,20]. Therefore, the ongoing rise in multidrug-resistant fungi such as *C. auris* demands continued vigilance and further efforts to develop novel antifungal agents and treatment strategies.

Epigenetics refers to stable and heritable changes in gene expression that are not due to alterations in the DNA sequence but are regulated through post-translational modifications (PTMs) of histones, DNA methylation, and RNA-based mechanisms. Histone PTMs play a pivotal role in modulating chromatin structure, thereby influencing DNA accessibility[21,22]. Proteins involved in this process are categorized as "writers," "readers," or "erasers," functioning to interpret and regulate the histone code through various PTM combinations[23,24]. Recent studies have demonstrated the critical role of histone H3 PTMs, such as methylation and acetylation, in driving both drug resistance and pathogenicity in human fungal pathogens. For instance, the histone H3 acetyltransferase complex SAGA is essential for the pathogenicity of fungi such as *C. albicans*, *Nakaseomyces glabrata* (formerly known as *Candida glabrata*), *C. neoformans*, and *A. fumigatus*[25–28]. Disrupting the gene encoding the acetyltransferase catalytic subunit Gcn5 results in significant defects in the pathogenicity and drug resistance of *C. albicans* and *N. glabrata*, as well as reduced pathogenicity in *C. neoformans* and multiple virulence-related phenotypes in *A. fumigatus*. Furthermore, the histone H3K56 acetyltransferase Rtt109 has been linked to the pathogenicity of *C. albicans* and its resistance to echinocandins and azole drugs[29,30]. The COMPASS complex, which includes the histone H3K4 methyltransferase catalytic subunit Set1, promotes resistance in *N. glabrata* and *Saccharomyces cerevisiae* to azole drugs and brefeldin A by regulating genes involved in the ergosterol biosynthesis pathway and drug efflux pumps[31,32]. Additionally, Set1 significantly influences *C. albicans* virulence in mouse models

of invasive infections[33,34]. The inhibitor MGCD290 is a new type of antifungal drug targeting the histone deacetylase Hos2 and currently in Phase II clinical trials[35], highlighting the therapeutic potential of histone H3 PTMs as antifungal targets. However, the role of histone H3 PTMs in the pathogenicity and drug resistance of *C. auris* remains unexplored.

In this study, we systematically investigated the effects of histone H3 PTM writers, including the acetyltransferases Gcn5 and Rtt109 and the methyltransferases Set1, Set2, and Dot1, on the pathogenicity and drug resistance of *C. auris*. Our research demonstrates that the disruption of genes encoding these histone H3 PTM writers leads to varied susceptibilities to a range of antifungal agents. Specifically, the histone H3 acetyltransferase Gcn5 is essential for the resistance of *C. auris* to azole, echinocandin, and polyene antifungal drugs, as well as for its virulence. Additionally, we found that the Gcn5-specific inhibitor CPTH$_2$, when combined with the echinocandin antifungal agent caspofungin, exhibits a synergistic effect against *C. auris* in both in vitro and in vivo models. These findings suggest that targeting epigenetic regulators such as Gcn5 could represent a promising strategy to address the challenges of drug resistance posed by this emerging superbug.

## Results

### Histone H3 PTMs in *C. auris* exhibit dynamic responses to antifungal exposure

In our initial foray for the relationship between histone H3 PTMs and antifungal resistance in *C. auris*, we treated the fluconazole-resistant strain CBS12767 (Clade I) with various antifungals, including the echinocandin caspofungin (CAS), the azole fluconazole (FLC), and the pyrimidine analog 5-fluorocytosine (5-FC). We then examined the global patterns of histone modification changes in acetylation and methylation. Interestingly, compared to the control (DMSO treatment), CAS treatment significantly increased acetylation at specific lysine residues on histone H3, particularly at K9, K14, K18, K27, K36, and K56 (Fig. 1a). Concurrently, CAS exposure also led to an elevated level of trimethylation at histone H3K36 (Fig. 1b), highlighting a potential epigenetic mechanism underlying the emergence of antifungal drug resistance in *C. auris*. Conversely, treatments with FLC or 5-FC did not induce noticeable changes in these histone modifications (Fig. 1a, b). Further multiple sequence alignment analysis, comparing the genome sequences of model organisms *S. cerevisiae* and *C. albicans*, allowed us to identify specific gene homologs in *C. auris* responsible for these histone modifications. These included acetyltransferases encoded by *B9J08_005133* (*GCN5*) and *B9J08_000852* (*RTT109*), as well as methyltransferases encoded by *B9J08_005356* (*SET1*), *B9J08_003450* (*SET2*), and *B9J08_000044* (*DOT1*).

Next, we exposed both the CBS12767 (Fig. 1c) and an echinocandin-resistant strain yCB799 (Clade I) (Fig. 1d) to CAS and FLC, respectively, monitoring their influences on the expression levels of the genes mentioned above. The treatments notably upregulated the expression of *GCN5* and *SET2* genes in both strains. Additionally, CAS treatment specifically triggered a significant increase in *SET1* expression in CBS12767, while the expressions of *RTT109* and *DOT1* remained largely unchanged (Fig. 1c, d). These findings collectively emphasize the sensitivity of histone H3 PTMs, specifically acetylation and methylation in *C. auris*, to antifungal treatment. This highlights the complex interactions between epigenetic regulation and drug resistance in this emerging fungal pathogen.

### The genes encoding histone-modifying enzymes are functionally conserved in *C. auris*

To investigate the impact of histone H3 PTMs on drug resistance and pathogenic traits in *C. auris*, we employed a genetic approach targeting the CBS12767 strain to knock out genes responsible for histone H3 modifications, specifically those encoding the enzymes *GCN5*, *RTT109*,

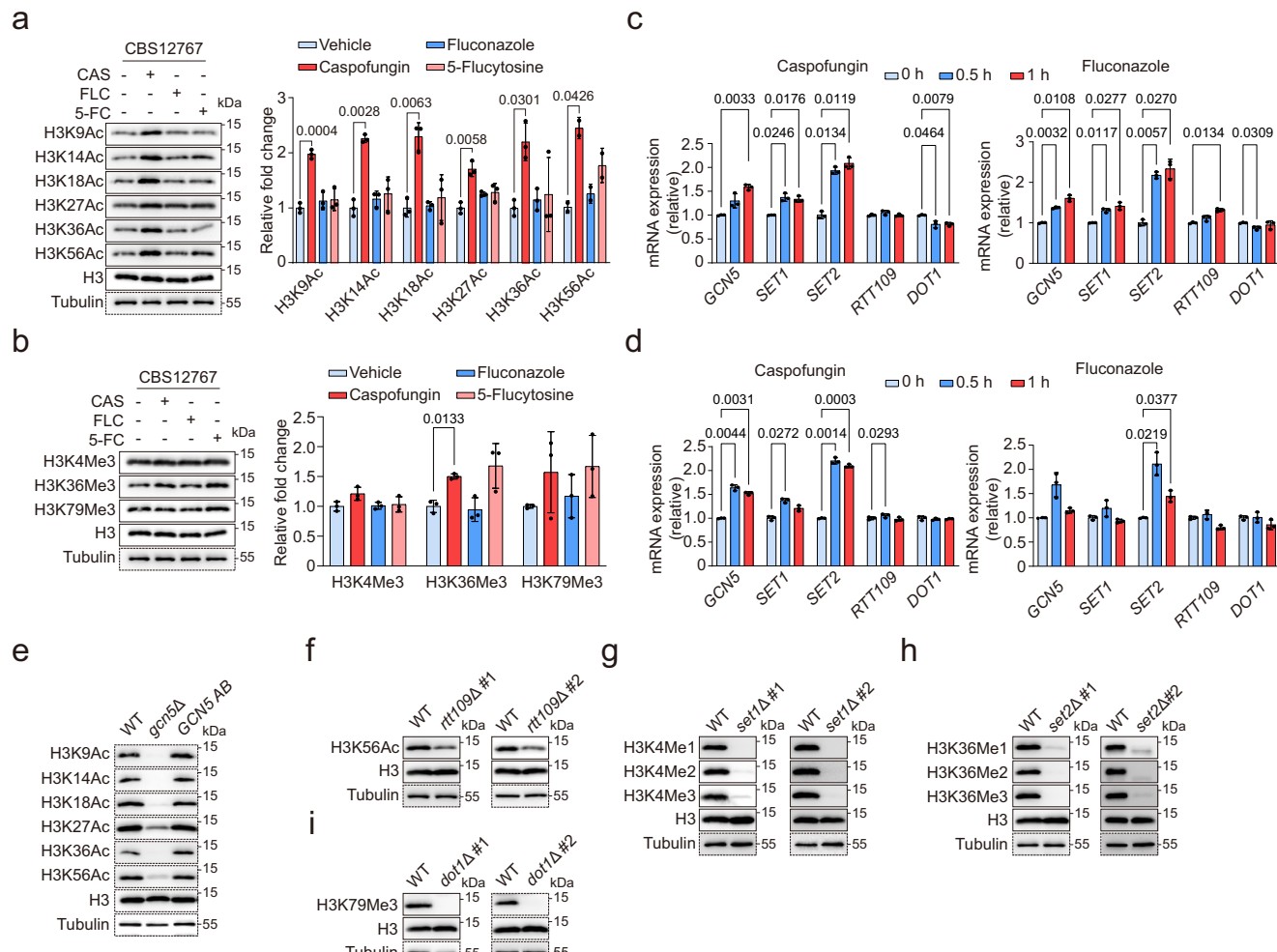

**Fig. 1 | Methylation and acetylation of histone H3 in *C. auris* respond to antifungal drug stress. a, b** Treatment of *C. auris* fluconazole-resistant strain CBS12767 with 100 ng/mL CAS (caspofungin), 128 μg/mL FLC (fluconazole), 8 μg/mL 5-FC (5-fluorocytosine), or an equal volume of DMSO for 1 h. Western blot was used to detect changes in acetylation (**a**) and methylation (**b**) levels of histone H3. **c, d** Treatment of *C. auris* fluconazole-resistant strain CBS12767 (**c**) or echinocandin-resistant strain yCB799 (**d**) with 100 ng/mL or 8 μg/mL CAS, 128 μg/mL FLC, 8 μg/mL 5-FC, and an equal volume of DMSO for 0, 0.5, 1, and 2 h. qRT-PCR was used to detect changes in transcription levels of genes *GCN5*, *RTT109*, *SET1*, *SET2*, and *DOT1*.

**e-i** Western blot was used to detect changes in acetylation or methylation levels of histone H3 corresponding to each gene knockout strain and *GCN5*-complemented strain (*GCN5 AB*). Each experiment was independently repeated twice with consistent results. #1 and #2 denote two independently constructed knockout strains. Data presented in (**a-d**) are expressed as mean ± SD and are representative of two or three independent experiments. Statistical significance analysis was performed using one-way ANOVA with Sidak's test. Source data are provided as a Source Data file.

*SET1*, *SET2*, and *DOT1*. Here, a *GCN5* complementation strain (*GCN5 AB*) was constructed in the *gcn5Δ* background and each of *SET1*, *SET2*, *RTT109* and *DOT1* was independently deleted in duplicate (#1 and #2 mutants for each gene). Through Western blot analysis, we evaluated the conservation of these gene functions in *C. auris* by analyzing the levels of acetylation and methylation at various lysine residues on histone H3 in both wild-type and the indicated mutants (Fig. 1e–i). Our findings demonstrated that in the absence of *GCN5*, key acetylation marks on histone H3 at lysine positions K9, K14, K18, and K36 were nearly eliminated, with a significant reduction in acetylation at K27, underscoring the critical role of *GCN5* in these modifications. Conversely, the *GCN5 AB* strain exhibited no discernible differences in acetylation patterns compared to the wild-type CBS12767 strain (Fig. 1e), affirming the preserved function of *GCN5* in *C. auris*. Notably, while the *RTT109* knockout resulted in a marked decrease in H3K56 acetylation, this modification was not entirely lost, indicating a divergence from observations in *S. cerevisiae*[36] (Fig. 1f). In contrast, H3K56 acetylation was almost completely absent in *gcn5Δ*. This suggests a collaborative role between Gcn5 and Rtt109 in mediating H3K56

acetylation, with a more pronounced contribution from Gcn5 (Fig. 1e, f). Furthermore, our study revealed that deletions of *SET1* or *SET2* led to the complete loss of mono-, di-, and trimethylation at H3K4 or H3K36, respectively, and knocking out *DOT1* resulted in the absence of H3K79 trimethylation (Fig. 1g–i). In summary, our data indicate that the homologous genes encoding histone-modifying enzymes, such as Gcn5, Rtt109, Set1, Set2, and Dot1, are highly conserved in *C. auris*, and their products retain the ability to modify chromatin structure. This reinforces the concept of a conserved epigenetic regulatory framework within this pathogen.

### Antifungal resistance in *C. auris* is linked to the activity of Lysine acetyltransferase Gcn5

To evaluate the impact of histone H3 PTMs on antifungal resistance in *C. auris*, we performed a comparative analysis of drug sensitivity among wild-type and mutant strains using a range of clinically relevant antifungals. Drug susceptibility tests using spot assays revealed that the isogenic mutant strains lacking *GCN5*, specifically, *gcn5Δ* from the *C. auris* CBS12767 background and *gcn5Δ-2* from the yCB799

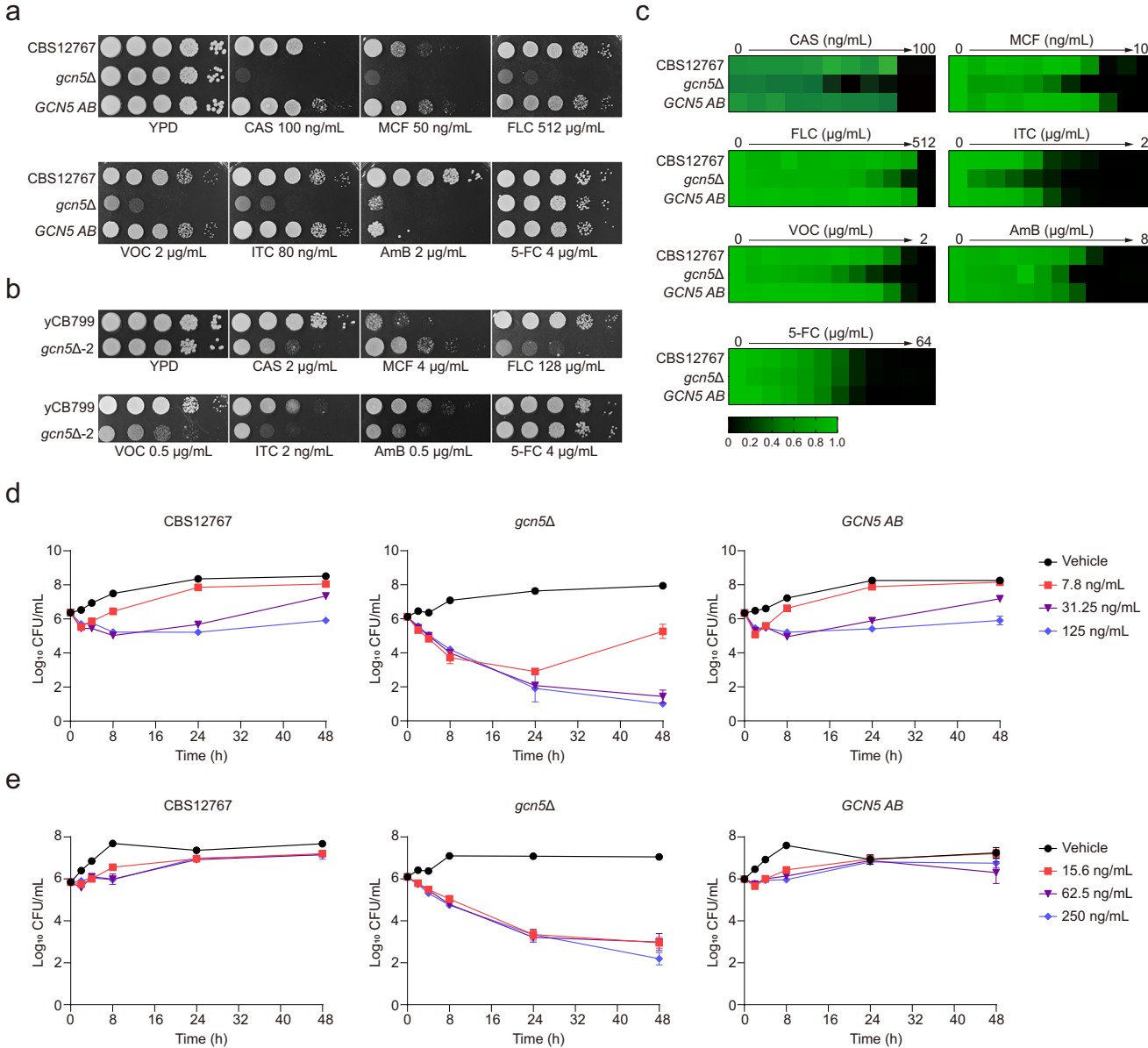

**Fig. 2 | Significant impact on antifungal drug resistance in *C auris* due to disruption of the *GCN5* gene encoding histone H3 acetyltransferase. a, b** Spot assay comparing the growth differences of *C. auris* CBS12767, *gcn5Δ*, and *GCN5 AB* (**a**) or yCB799 and *gcn5Δ-2* (**b**) under antifungal drug stress with CAS (caspofungin), MCF (micafungin), FLC (fluconazole), VOC (voriconazole), ITC (itraconazole), AmB (amphotericin B), or 5-FC (5-fluorocytosine). **c** Microdilution method to test the effects of antifungal drugs CAS, MCF, FLC, VOC, ITC, AmB, or 5-FC in YPD liquid medium on the growth of *C. auris* CBS12767, *gcn5Δ*, and *GCN5 AB*. **d, e** Time-kill curves of CAS (**d**) or MCF (**e**) in YPD liquid medium on *C. auris* CBS12767, *gcn5Δ*, and *GCN5 AB* strains. Data presented in (**d, e**) are expressed as mean ± SD and are representative of three independent experiments. Source data are provided as a Source Data file.

background, exhibited significantly increased sensitivity to various antifungal classes, including echinocandins, azoles, and polyenes (Fig. 2a, b). The *GCN5 AB* strain, constructed in the *gcn5Δ* background, maintained antifungal sensitivity levels similar to those of the wild-type strain, with the exception of amphotericin B (AmB). This discrepancy may be attributed to disruptions in the *LEU2* gene resulting from the integration of the *GCN5* ORF (Fig. 2a), as we observed that the *LEU2* knockout strain (*leu2Δ*) in the CBS12767 background showed increased sensitivity to AmB while retaining wild-type resistance to other antifungals (Fig. S1c). Crucially, we found that when cells were treated with CPTH₂ at a concentration of 0.5 μM, a concentration that effectively inhibits Gcn5-mediated histone H3 acetylation, both wild-type and *leu2Δ* mutant strains displayed similar drug sensitivity profiles (Fig. S1d, e). This outcome strongly suggests that the lack of *LEU2*

has no impact on the drug response mediated by Gcn5. Confirmatory evidence was obtained by creating a double knockout strain (*leu2Δgcn5Δ*) through deletion of both *LEU2* and *GCN5*, which showed no measurable changes in drug sensitivity compared to the *gcn5Δ* mutant (Fig. S1f). Interestingly, we observed a differential response to micafungin (MCF) among the knockout strains: *gcn5Δ* displayed heightened sensitivity, while *gcn5Δ-2* demonstrated increased resistance, likely due to variations in the gene expression and/or genetic backgrounds of the two parental strains (Fig. 2a, b). Deletion of *GCN5* in both wild-type strains did not significantly affect their resistance to 5-FC (Fig. 2a, b). Additionally, mutants lacking the *SET1* and *RTT109* genes in the CBS12767 background exhibited enhanced vulnerability to echinocandins, azoles, and AmB, although the effects were less pronounced compared to the *gcn5Δ* strain (Fig. S1a). The deletion of

*DOT1* resulted in increased sensitivity to AmB and a moderate elevation in susceptibility to echinocandins. In contrast, the azole susceptibilities of the deletion mutant remained comparable to those of the wild-type strain CBS12767 (Fig. S1a). Furthermore, the *set2Δ* mutant showed increased resistance to azole antifungals, implicating Set2 as a negative regulator of azole resistance in *C. auris*, in line with observations in *N. glabratus*[37] (Fig. S1a). Growth curve analyses under antifungal pressure corroborated the essential role of Gcn5 in maintaining the viability of *C. auris*, as the *gcn5Δ* strain exhibited near-total growth inhibition (Fig. S1b). A standard broth microdilution sensitivity test yielded consistent results, demonstrating that *gcn5Δ* mutant strains exhibited heightened susceptibility to echinocandins, azoles, and AmB relative to the wild-type strain (Fig. 2b). Minimum inhibitory concentrations (MICs) for antifungal drugs were assessed in both wild-type and indicated mutant strains following Clinical and Laboratory Standards Institute (CLSI) guidelines (CLSI M27-A4), with detailed outcomes summarized in Table S1. To further examine the role of Gcn5 in regulating the response of *C. auris* to the fungicidal activity of echinocandin, we performed time-kill assays, which revealed that both CAS and MCF treatments significantly enhanced fungicidal effects on *gcn5Δ* strains (Fig. 2d, e), illustrating the importance of Gcn5 in modulating echinocandin resistance.

In summary, our findings highlight the critical role of histone H3 PTM writers in shaping *C. auris*'s drug resistance profile. Given that the deletion of *GCN5* in *C. auris* produced the most pronounced responses to antifungals, we concentrated our subsequent experiments on this gene to explore its functions and underlying mechanisms.

## Histone acetyltransferase Gcn5 is essential for azole resistance by regulating drug-responsive gene expression

The modification of histones by acetyl groups is critical in regulating chromatin structure and transcription. Histone acetyltransferase Gcn5 has been previously recognized as a positive regulator of gene expression through its interaction with target promoters[38]. To investigate its mode of action at a genome-wide level, we employed RNA-seq technology to analyze the Gcn5 transcriptome in *C. auris* treated with or without FLC. Both WT and *gcn5Δ* mutant strains were treated with 256 μg/mL FLC or an equivalent volume of DMSO for 2 h during the early exponential growth phase, followed by RNA extraction and sequencing. Our results revealed significant gene expression shifts in the *gcn5Δ* strain compared to WT, with a total of 975 differentially expressed genes (DEGs) identified, including 517 upregulated and 458 downregulated genes (Fig. 3a). Interestingly, after 2 h of FLC treatment, genes involved in ergosterol biosynthesis and drug efflux pumps were significantly upregulated in the WT strain but not in the *gcn5Δ* strain (Fig. 3b). It is noteworthy that several genes, including *FCR1*, previously identified as a resistance inhibitor[39], were upregulated in the *gcn5Δ* strain following FLC treatment (Fig. 3c). Gene set enrichment analysis (GSEA) of the DEGs further highlighted the ergosterol biosynthesis pathway (Fig. 3c). Quantitative PCR validation of the sequencing data confirmed that FLC treatment markedly upregulated the expression of genes associated with ergosterol biosynthesis (*ERG11, ERG1, ERG3, ERG25*) and the transcription factor *UPC2*, as well as genes for ABC superfamily drug efflux pumps (*CDR1, SNQ2*) and the MFS superfamily drug efflux pump (*MDR1*) in the WT strain CBS12767. In contrast, the expression of these genes was significantly attenuated in the *gcn5Δ* strain, with levels restored to near WT levels in the *GCN5 AB* strain (Fig. 3d), a consistency observed in both yCB799 and *gcn5Δ-2* strains (Fig. S2). Furthermore, the absence of FLC treatment led to a notable reduction in the expression of specific genes, such as *ERG11* and *ERG25*, in the *gcn5Δ* strain, underscoring the essential role of Gcn5 in maintaining basal gene expression.

To ascertain the influence of Gcn5-mediated histone H3 acetylation on the expression of genes linked to azole resistance, we employed ChIP-qPCR to assess the enrichment of acetylated H3K14 (H3K14Ac) at the promoter regions of the azole target gene *ERG11* and the azole efflux pump gene *CDR1*. As expected, we observed a pronounced accumulation of H3K14Ac in the regions upstream of the ORF of *ERG11* and *CDR1* in the WT strain. This phenomenon intensified 30 min post-FLC administration (Fig. 3e, f). In contrast, this accumulation was significantly diminished in the *gcn5Δ* variant (Fig. 3e, f), highlighting the critical role of Gcn5 in modulating the expression of *ERG11* and *CDR1* through a specific histone H3 acetylation mechanism. Additionally, we compared the functional efficacy of drug efflux systems in WT and *gcn5Δ* mutants. Results from a Rhodamine 6G efflux assay demonstrated that the *gcn5Δ* mutant exhibited a significant defect in expelling Rhodamine 6G from the cells, in stark contrast to the parent CBS12767 strain, which efficiently removed the dye (Fig. 3g, h). This indicates the indispensable role of Gcn5 in maintaining the operational integrity of drug efflux systems in *C. auris*.

## Gcn5 is required for echinocandin resistance by regulating the expression of genes involved in calcineurin signaling pathway and cell wall integrity

To further investigate the mechanism by which Gcn5 regulates echinocandin drug resistance in *C. auris*, we first quantified the expression levels of echinocandin target genes, specifically those encoding the β-1,3-D-glucan synthase catalytic subunits *FKS1* and *FKS2*, as well as the regulatory subunit *RHO1*[40,41], across various *C. auris* strains using qRT-PCR. Markedly, we observed divergent expression patterns: both *FKS1* and *RHO1* expression were decreased, whereas *FKS2* was increased in the *gcn5Δ* and *gcn5Δ-2* strains (Fig. 4a, Fig. S3a). This finding suggests a potential compensatory mechanism, given the functional redundancy between *FKS1* and *FKS2*[42–44]. Such complexity indicates that the role of Gcn5 in echinocandin resistance may extend beyond mere regulation of *FKS* gene expression.

The highly conserved calcineurin signaling pathway is crucial for maintaining calcium ion homeostasis within fungi, a crucial aspect of their survival and adaptation[45]. Importantly, this pathway has been implicated in cell wall integrity and echinocandin resistance[46–48]. We therefore hypothesized that Gcn5-dependent echinocandin resistance may involve the activation of the calcineurin signaling pathway. To test this hypothesis, we first examined the impact of Gcn5 on the transcriptional levels of genes associated with this pathway. Calcineurin is a broadly conserved protein phosphatase comprising a catalytic A subunit and a Ca²⁺-binding regulatory B subunit, both essential for its enzymatic function[49]. The zinc finger transcription factor Crz1 has been identified as a target of calcineurin in yeast[50]. Moreover, the transcription factor Cas5 acts as a regulator of resistance to the echinocandin caspofungin in *C. albicans*, with studies showing that genes regulated by Cas5 in response to cell wall stress are also linked to calcineurin[48,51,52]. *GLC7* encodes a phosphatase critical for activating Cas5 via dephosphorylation, facilitating its entry into the nucleus[48]. Our results demonstrated that CAS exposure significantly increased the expression of *CRZ1* and *CAS5* in the wild-type strain; however, this induction was diminished in mutants lacking *GCN5* (Fig. 4b, Fig. S3b). Additionally, *GLC7* expression was notably reduced in the *gcn5Δ* mutant compared to the wild-type (Fig. S3b, c). Consistently, deletion of the gene encoding the calcineurin catalytic subunit (*CNA1*) or the transcription factor *CAS5* in both *C. auris* CBS12767 and yCB799 backgrounds resulted in enhanced sensitivity to CAS, a phenotype mirroring that of *gcn5Δ* mutants (Fig. 4c). This supports a potential role for the calcineurin pathway in Gcn5-dependent echinocandin resistance. Additionally, it has been reported that Gcn5-deficient mutants in both *N. glabratus* and *C. neoformans* exhibit significantly heightened sensitivity to the calcineurin inhibitor FK506[53,54]. Furthermore, Gcn5 regulates the expression of *CNA1* in *C. neoformans*[55], further bolstering our conclusions. Unexpectedly, we found that the sensitivity of the *crz1Δ* mutant to CAS was comparable to that of the Wild-type (Fig. 4c). Due to the challenges associated with genetic

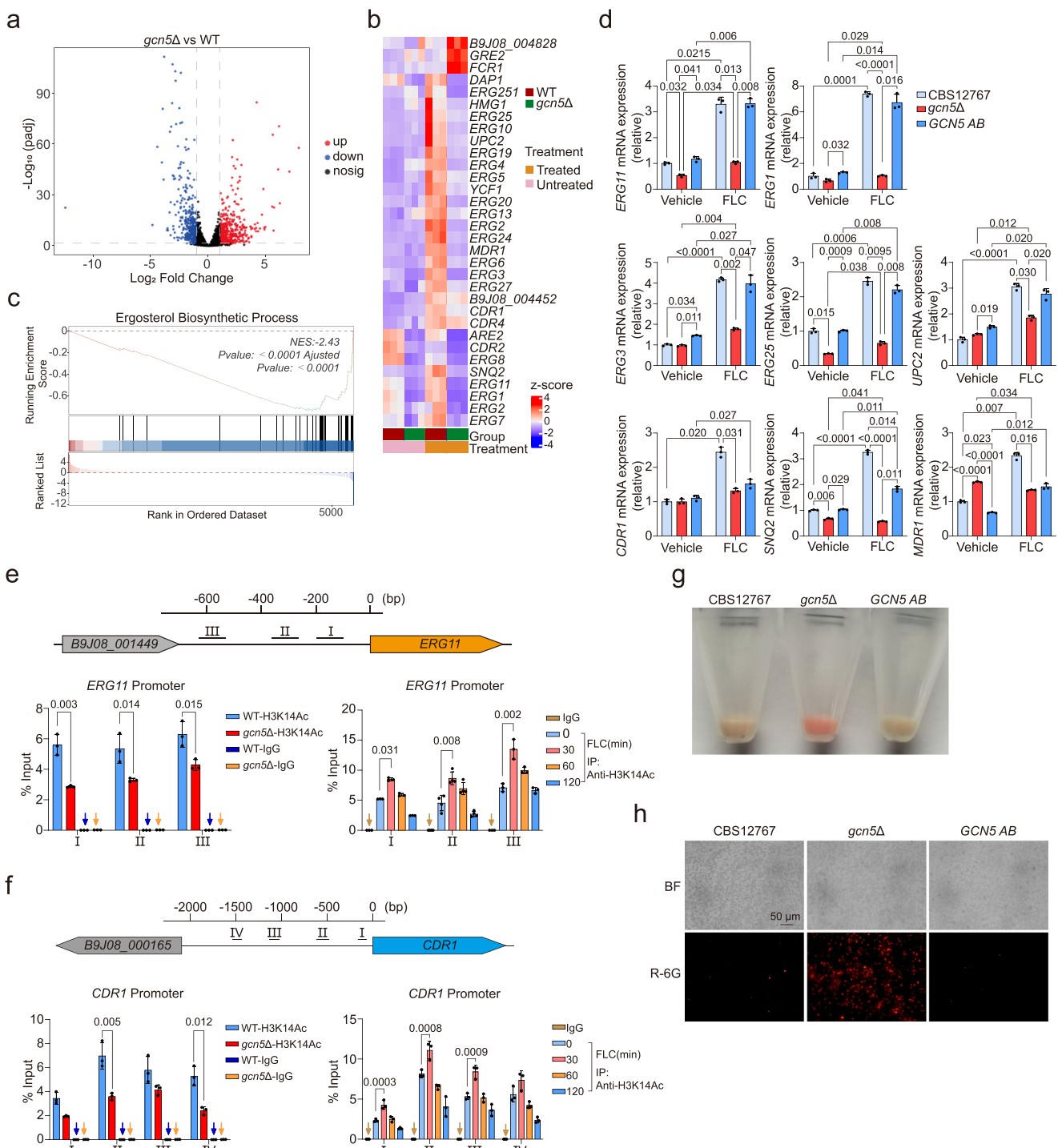

**Fig. 3 | Gcn5 regulates both ergosterol synthesis and drug efflux pump activity in *C. auris*. a-d** Transcriptomic sequencing of *C. auris* CBS12767 or *gcn5Δ* treated with 256 μg/mL FLC (fluconazole) or an equal volume of DMSO for 2 h. **a** Volcano plot showing differentially expressed genes between *gcn5Δ* and CBS12767. Statistical analysis was performed using DESeq2 (two-sided Wald test), with multiple testing correction via the Benjamini–Hochberg method. **b** Heatmap illustrating expression changes in genes related to ergosterol biosynthesis and drug efflux pumps. **c** GSEA analysis revealed significant enrichment of the ergosterol biosynthesis pathway in differentially expressed genes between *gcn5Δ* and CBS12767. Statistical significance was determined using a two-sided permutation test (1000 permutations). The normalized enrichment score (NES) was 2.43, with both nominal and FDR-adjusted *p*-values < 0.0001. **d** qRT-PCR validation of selected genes involved in ergosterol synthesis (*ERG11, ERG1, ERG3, ERG25, UPC2*) and drug efflux (*CDR1, SNQ2, MDR1*) in CBS12767, *gcn5Δ*, and *GCN5 AB*. **e, f** Gcn5-dependent

H3K14Ac at the promoter regions of *ERG11* and *CDR1* assessed by ChIP-qPCR. **e** Schematic of H3K14Ac enrichment at different positions in the *ERG11* promoter (top); Detection of H3K14Ac enrichment in the *ERG11* gene promoter region of *C. auris* CBS12767 or *gcn5Δ* (bottom left); Fluconazole-induced changes in H3K14Ac levels at the *ERG11* promoter of *C. auris* CBS12767 over time (30, 60, 120 min) (bottom right). **f** Similar analysis for the *CDR1* promoter. **g, h** R-6G (Rhodamine 6-G) efflux assay testing drug efflux pump activity in *C. auris* CBS12767, *gcn5Δ*, and *GCN5 AB*, with consistent results across two independent repeats. **g** shows photos of pellets after starvation, dye uptake, glucose stimulation, and centrifugation. **h** presents the same samples viewed under fluorescence microscopy in the RFP channel. Data presented in (**d-f**) are expressed as mean ± SD and are representative of three independent experiments. Statistical significance analysis was performed using two-way ANOVA (**d**) or one-way ANOVA (**e, f**) with Sidak's test. Source data are provided as a Source Data file.

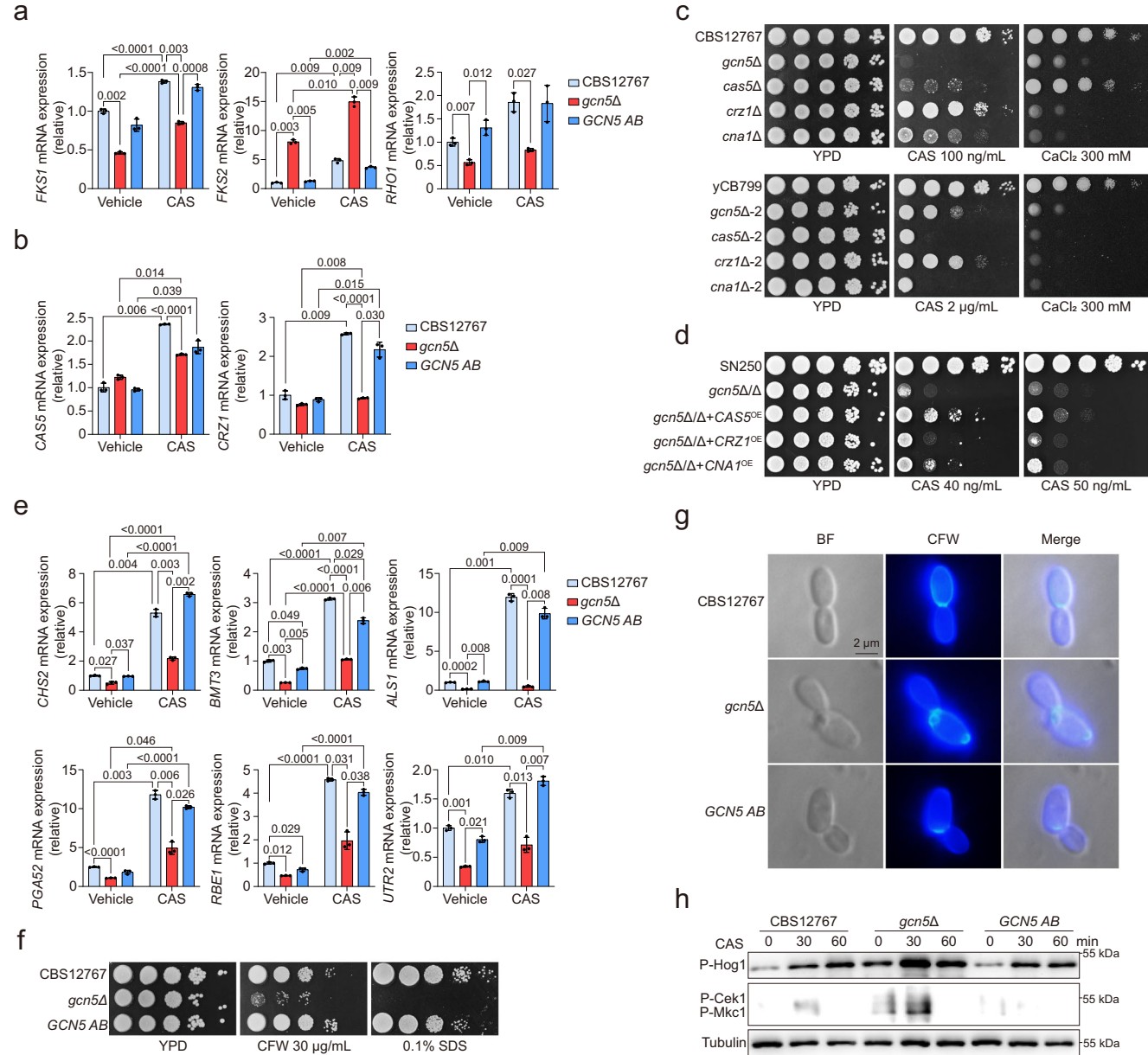

**Fig. 4 | Gcn5 regulates echinocandin drug resistance and cell wall structure homeostasis in *C. auris* by modulating the calcineurin pathway and the transcription factor Cas5. a**, **b** qRT-PCR was used to measure expression levels of the *FKS1*, *FKS2* and *RHO1* genes (**a**) and the *CAS5* and *CRZ1* genes (**b**) in *C. auris* strains CBS12767, *gcn5Δ*, and *GCN5 AB*. **c** Spot assay comparing growth differences of *C. auris* CBS12767, *gcn5Δ*, *cas5Δ*, *crz1Δ*, and *cna1Δ* (top) or *C. auris* yCB799, *gcn5Δ-2*, *cas5Δ-2*, *crz1Δ-2*, and *cna1Δ-2* (bottom) under 100 ng/mL or 2 μg/mL CAS and 300 mM CaCl₂ conditions. **d** Spot assay assessing growth differences of *C. albicans* SN250, *gcn5Δ/Δ*, *gcn5Δ/Δ+CAS5*ᴼᴱ, *gcn5Δ/Δ+CNA1*ᴼᴱ and *gcn5Δ/Δ+CRZ1*ᴼᴱ under 40 ng/mL or 50 ng/mL CAS (caspofungin) conditions. **e** qRT-PCR analysis of expression changes in genes related to fungal cell wall integrity regulated by the transcription factor Cas5. **f** Spot assay detecting growth differences of *C. auris*

CBS12767, *gcn5Δ*, and *GCN5 AB* strains under cell wall stress conditions of 30 μg/mL CFW (Calcofluor White) or 0.1% W/V SDS (Sodium Dodecyl Sulfate). **g** Fluorescence microscopy using the UW channel to observe chitin distribution in the cell walls of *C. auris* CBS12767, *gcn5Δ*, and *GCN5 AB* after staining with CFW. The experiment was independently repeated twice with consistent results. **h** Western blot analysis of phosphorylation levels of MAPK pathway kinases Hog1, Mkc1, and Cek1 in *C. auris* CBS12767, *gcn5Δ*, and *GCN5 AB* growth under normal conditions or after treatment with 100 ng/mL CAS for 30 or 60 min. The experiment was independently repeated twice with consistent results. Data presented in (**a–e**) are expressed as mean ± SD and represent of three independent experiments. Statistical significance analysis was performed using two-way ANOVA with Sidak's test. Source data are provided as a Source Data file.

manipulation in *C. auris*, we explored an alternative approach by attempting to overexpress *CAS5*, *CNA1* or *CRZ1* in the *C. albicans gcn5Δ/Δ* mutant. Interestingly, overexpression of *CAS5* and *CNA1*, but not *CRZ1*, partially restored resistance to CAS in the *gcn5Δ/Δ* mutant (Fig. 4d). Furthermore, to clarify the regulatory relationship between Gcn5 and Cna1, we examined whether *CNA1* deletion affects Gcn5-dependent histone H3 acetylation. We found that the *cna1Δ* strain exhibited no significant changes in Gcn5-mediated H3 acetylation

levels compared to the wild-type, regardless of caspofungin treatment (Fig. S3d). These findings strongly suggest that the influence of the calcineurin pathway on Gcn5-mediated echinocandin resistance occurs independently of Crz1, potentially functioning through Cas5 instead, thereby highlighting a complex regulatory pathway distinct from previously characterized models.

The observation that Gcn5 impacts the expression of genes such as *CRZ1* and *CAS5* led us to hypothesize that this histone acetyltransferase

may play a critical role in modulating the calcium-calcineurin signaling pathway and cell wall integrity stress. As illustrated in (Fig. 4c), the *C. auris* mutant lacking *GCN5* exhibited increased susceptibility to elevated calcium ion levels, akin to the phenotypes observed with the knockout of *CNA1*, *CRZ1*, or *CAS5*. This highlights the connection of Gcn5 to the calcineurin signaling pathway. Compared to the Wild-type, the *gcn5Δ* mutant displayed decreased expression levels of a range of genes (e.g., *CHS2*, *PGAS2*, *RBE1*, and *UTR2*) integral to cell wall stability and regulated by Cas5 under CAS treatment[48] (Fig. 4e, Fig. S4a). An interesting observation emerged from our analysis: neither CBS12767 nor yCB799 treated with MCF showed any significant changes in the expression of Cas5-regulated genes, particularly *CHS2*, *ALS1*, *PGA42*, and *RBE1*, which were significantly upregulated under CAS treatment. This finding implies that MCF and CAS may elicit distinct stress response pathways in *C. auris* and importantly, the differential role of Gcn5 in modulating susceptibility to MCF is unlikely to be attributable to transcriptional disparities of these already characterized Gcn5-regulated genes between CBS12767 and yCB799 (Fig. S4b-e). Genomic analysis revealed mutations (S639F and L906S) in the *FKS1* locus exclusively in strain yCB799, while strain CBS12767 remained wild-type at this locus (Fig. S4f), suggesting that these mutations potentially contribute to the strain-specific differences in MCF sensitivity, a hypothesis that warrants further functional validation. Additionally, defects in cell wall stability following exposure to Calcofluor White (CFW) or Sodium Dodecyl Sulfate (SDS) were observed in the *gcn5Δ* mutant (Fig. 4f). In particular, we observed marked disorganization in chitin distribution, particularly around the septum, in the cell wall of the *gcn5Δ* mutant, in contrast to the orderly arrangement in the WT and *GCN5 AB* strains (Fig. 4g). Consistently, phosphorylation of the Hog1, Mkc1, and Cek1 MAP kinases was significantly elevated in the *gcn5Δ* mutant compared to the wild-type, suggesting that the absence of *GCN5* induces cell wall perturbations in *C. auris* (Fig. 4h). These results suggest that this complex interaction not only highlights the critical role of Gcn5 in echinocandin resistance and cell wall stability but also reveals the sophisticated regulatory mechanisms fungi utilize to adapt to environmental stressors and antifungal pressures. To further validate the direct role of Gcn5 in regulating calcineurin signaling and *FKS1* expression through histone H3 acetylation, we performed ChIP-qPCR to examine H3K14Ac enrichment at their promoter regions following CAS induction. Time-course experiments revealed progressive H3K14 acetylation at the *CASS*, *CRZ1*, and *FKS1* promoters, peaking ~1 h post-treatment—coinciding with the transcriptional activation of these stress-responsive genes (Fig. S4g–i).

In summary, our data emphasize the essential role of Gcn5 in maintaining the cell wall integrity of *C. auris*, particularly through its regulation of the calcineurin pathway and Cas5. This modulation influences the fungal response to echinocandin drugs and cell wall disturbances, illustrating the nuanced regulatory framework that enables fungal adaptation to environmental challenges and therapeutic interventions.

## Gcn5 is required for the pathogenicity of *C. auris*

Previous studies in human fungal pathogens such as *C. albicans* and *C. neoformans* have established a strong correlation between Gcn5 activity and fungal virulence[25,53,55], raising the question of its role in regulating the virulence of *C. auris*. To investigate this, we employed the *Galleria mellonella* infection model to compare survival and fungal burden post-infection between the wild-type CBS12767 and *gcn5Δ* mutant strains. Kaplan-Meier survival analysis revealed that the *gcn5Δ* mutant exhibited reduced virulence compared to the wild-type strain (Fig. S5a). Similarly, fungal burden analysis at 24 and 48 h post-infection showed a marked reduction in larvae infected with the *gcn5Δ* mutant strain compared to those infected with the wild-type strain (Fig. S5b). This virulence defect was further validated using an immunosuppressed mouse model of disseminated infection, as previously described[56]. Both survival analysis (Fig. 5a, b) and organ fungal burden

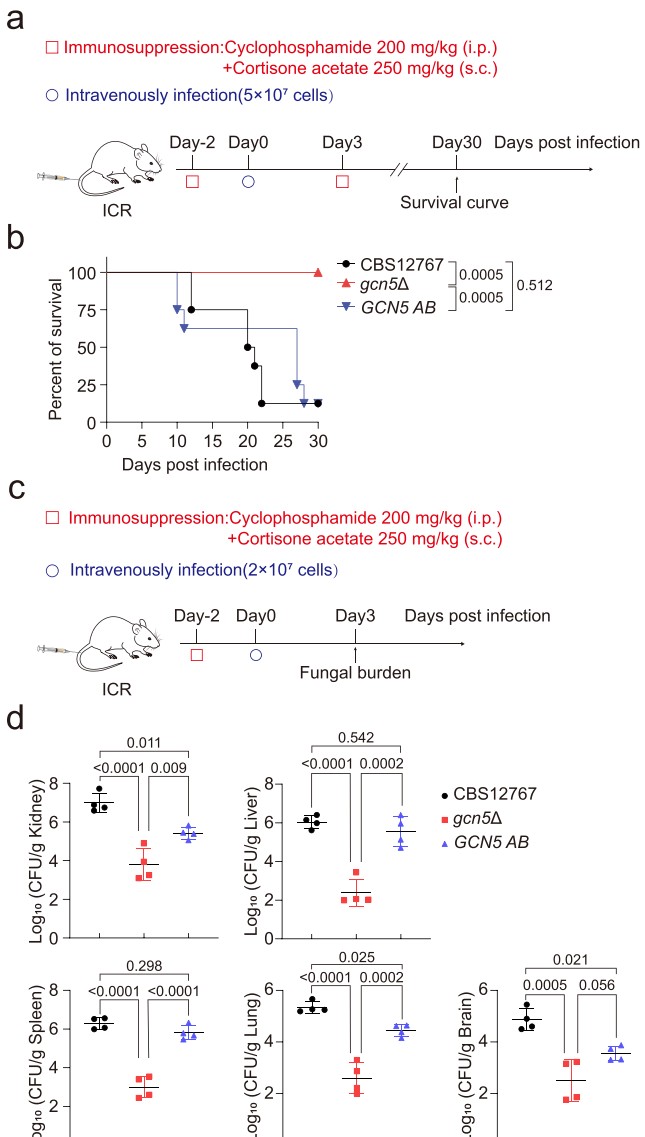

**Fig. 5 | Gcn5 is essential for the virulence of *C. auris* in invasive infections in mice. a** Schematic diagram outlining the procedure for generating survival curves. **b** Thirty-day survival curve of immunosuppressed ICR mice (*n* = 8) after intravenous infection with *C. auris* CBS12767, *gcn5Δ*, or *GCN5 AB*, monitored daily for mortality. **c** Schematic diagram illustrating the method for measuring fungal burden in organs. **d** Assessment of fungal burden in the kidneys, spleen, liver, brain, and lungs of immunosuppressed ICR mice (*n* = 4) 3 days post-infection with *C. auris* CBS12767, *gcn5Δ*, or *GCN5 AB*, quantified using CFU counts. Data presented in (**d**) are expressed as mean ± SD. Statistical significance analysis was performed using Log-rank (Mantel-Cox) test (**b**) or one-way ANOVA with Sidak's test (**d**). Source data are provided as a Source Data file.

measurements (Fig. 5c, d) demonstrated that immunosuppressed ICR mice infected with the Gcn5-deficient strain had a significantly higher survival rate and lower fungal burden across organs compared to the control group. These findings underscore the critical role of Gcn5 in the pathogenicity of *C. auris*.

## *C. auris gcn5Δ* mutant is more susceptible to clearance by antifungal agents in vivo

Our findings indicate that Gcn5 is essential for *C. auris* resistance to various antifungal agents and plays a significant role in its pathogenicity. Consequently, we propose that Gcn5 represents a promising

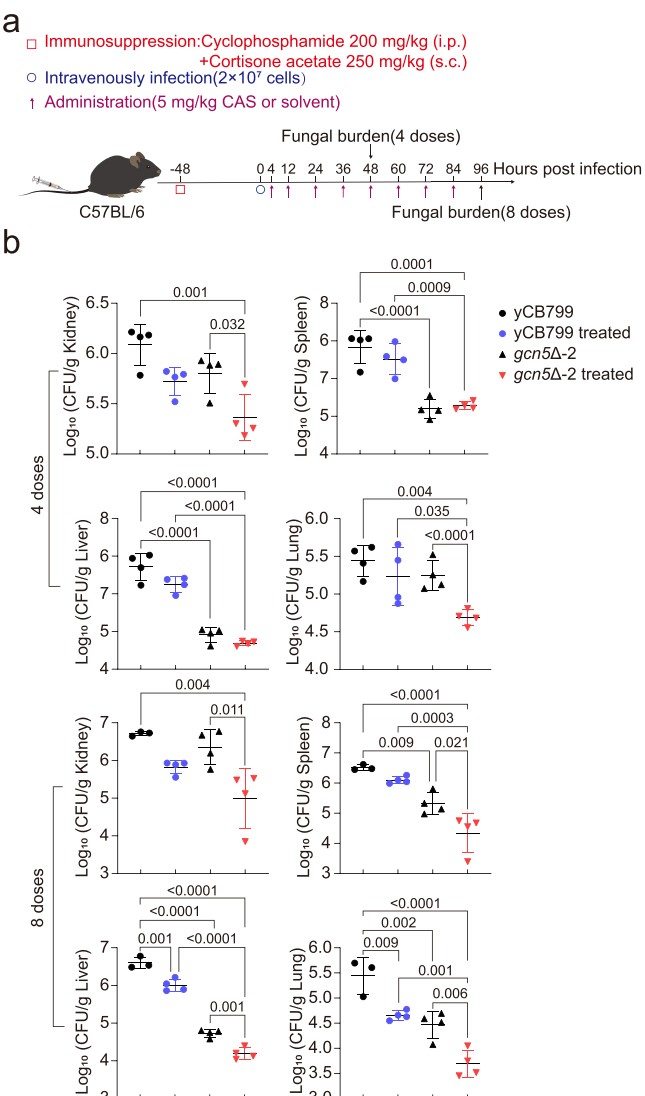

**Fig. 6 | The *gcn5*Δ mutant is more effectively cleared by CAS treatment in vivo. a** Schematic diagram outlining the experimental procedure. **b** Assessment of fungal burden in the kidneys, spleen, liver, and lungs of mice following treatment with 4 doses (top) or 8 doses (bottom) of CAS ($n = 4$). Fungal burden was measured using CFU counts. Data are expressed as mean ± SD. Statistical significance analysis was performed using one-way ANOVA with Sidak's test. Source data are provided as a Source Data file.

target for treating invasive infections of *C. auris*. To test this hypothesis further, we conducted experiments using immunocompromised C57BL/6 mice infected with either the echinocandin-resistant strain yCB799 or a *gcn5*Δ-2 mutant. Four hours post-infection, the mice were treated intraperitoneally with 5 mg/kg of CAS or a solvent control every 12 h, receiving either 4 or 8 doses. Following treatment, we assessed the fungal burden in the kidneys, liver, spleen, and lungs (Fig. 6a). Our results revealed that the fungal burden in these organs was significantly lower in mice infected with the *gcn5*Δ-2 mutant compared to those infected with the wild-type strain, particularly after 8 doses of CAS (Fig. 6b). In a separate experiment, we infected immunocompromised C57BL/6 mice with a fluconazole-resistant strain CBS12767 and its *gcn5*Δ mutant derivative. The mice received 20 mg/kg of fluconazole, administered 8 times following the same regimen as described above, alongside solvent control injections of equal volume. Fungal burden in various organs was measured 12 h after the final dose. Mice infected with the *gcn5*Δ mutant exhibited a

significant reduction in fungal burden compared to those infected with the wild-type strain in the untreated group, consistent with earlier findings (Fig. S6; Fig. 5d). Although not statistically significant, there was a notable trend toward reduced kidney fungal burden in *gcn5*Δ mutant-infected mice following fluconazole treatment (Fig. S6). These results suggest that the *gcn5*Δ mutant not only has reduced pathogenicity but is also more susceptible to CAS clearance. Thus, targeting Gcn5 in conjunction with antifungal therapy for *C. auris* invasive infections appears to be a highly effective strategy.

## Targeted inhibition of Gcn5 synergizes with CAS against *C. auris* in vitro and in vivo

Combination antifungal therapy has emerged as a promising strategy to overcome antifungal drug resistance and the high mortality rates associated with severe fungal infections[57]. Given the essential role of Gcn5 in regulating drug resistance and virulence in *C. auris*, along with the limited efficacy of currently available antifungals against multidrug-resistant strains, we explored the potential of combining Gcn5 inhibitors with antifungal medications to combat *C. auris* infections. Two specific Gcn5 inhibitors, CPTH$_2$ and MB-3[58–60], were evaluated. Treatment of *C. auris* cells with CPTH$_2$ at a concentration of 1 μM effectively inhibited the acetyltransferase activity of Gcn5, as evidenced by reduced acetylation levels of histone H3 at K14, K18, and K36 (Fig. 7a). While MB-3 was also tested, its efficacy was markedly lower than that of CPTH$_2$ (Fig. S7a), leading to its exclusion from further analysis.

We first assessed the combined antifungal activity of CPTH$_2$ with various antifungal agents against fluconazole-resistant (CBS12767) and echinocandin-resistant (yCB799) clinical isolates of *C. auris* (clade I) in solid culture medium. Remarkably, pairing CPTH$_2$ with the echinocandin CAS resulted in a significant reduction in fungal growth for both strains at dosages where neither agent alone had a noticeable effect (Fig. 7b). This combination exhibited a synergistic interaction, enhancing antifungal activity against both strains, as indicated by a Fractional Inhibitory Concentration Index (FICI) of <0.5 (Fig. 7c, d). Notably, this synergy could be translated into a more potent fungicidal effect (Fig. 7e–h), suggesting an improved therapeutic strategy against these formidable strains of *C. auris*. Consistent synergistic effects were observed across most clades of *C. auris*, with the exception of clade II (Fig. 7i, Fig. S7b, Table S2). Importantly, the combination of CPTH$_2$ and CAS failed to exhibit a synergistic effect on the *GCN5*-deficient mutant (Fig. 7j), confirming that the observed synergy is achieved through specific inhibition of Gcn5.

The synergistic effect of CPTH$_2$ and CAS against the echinocandin-resistant *C. auris* isolate was further investigated in vivo using the *Galleria mellonella* infection model. We found that a remarkably low dose of CPTH$_2$ (8 ng/mL) significantly improved survival rates in larvae infected with yCB799, demonstrating a dose-dependent effect (Fig. 7k). Furthermore, a combination of 16 ng/mL CPTH$_2$ and 0.5 μg/mL CAS led to a significant increase in survival rates compared to either agent used in isolation (Fig. 7k). When we measured the fungal burden at 24 and 48 h post-infection, we found that by 48 h, the treated groups (receiving CPTH$_2$, CAS, or a combination of both) exhibited a significant lower fungal burdens than the untreated control group. Notably, the combination treatment resulted in an even greater reduction in the fungal burden compared to using CAS alone, which further validates the enhanced antifungal efficacy of the combination therapy (Fig. S7c). We subsequently confirmed these findings in a mouse model, where C57BL/6 mice were infected with yCB799 and treated with CPTH$_2$ (10 mg/kg or 40 mg/kg), CAS (10 mg/kg), or both agents intraperitoneally every 12 h for a total of 6 doses, alongside a solvent control. Twelve hours after the final dose, we assessed the fungal burden in various organs. Results revealed that while CPTH$_2$ alone did not significantly affect fungal burden, the combination of CPTH$_2$ and CAS markedly reduced fungal burden in the kidneys and liver compared to CAS alone (Fig. 7l). However, the combination did

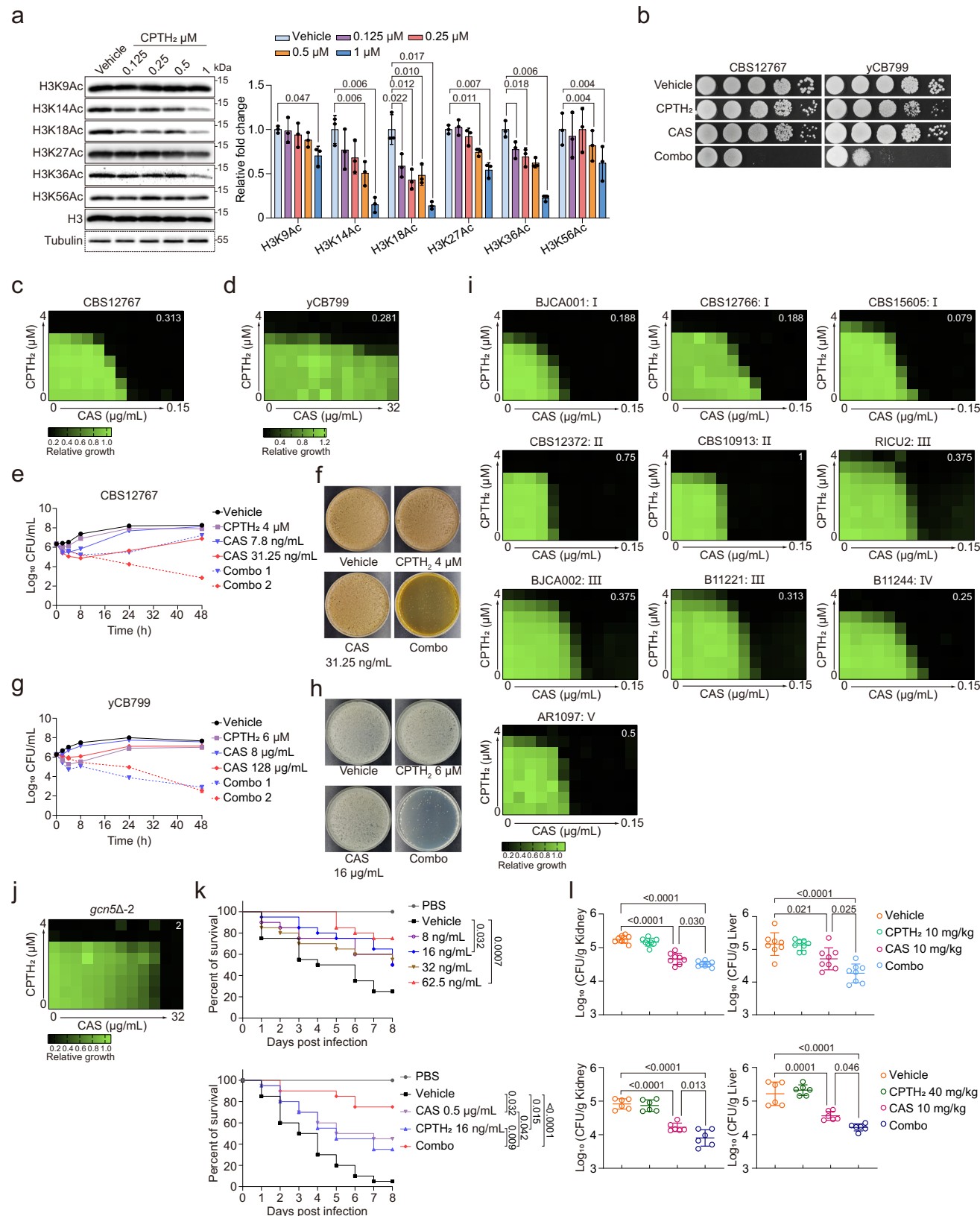

not significantly affect the spleen or lungs, nor did it improve the survival rate or survival time of immunosuppressed mice infected with yCB799 (Fig. S7d, e). These findings strongly indicate that the Gcn5 inhibitor CPTH2, in conjunction with CAS, offers a synergistic therapeutic strategy against invasive *C. auris* infections, highlighting its potential to augment antifungal efficacy.

A defining characteristic of *C. auris* is its ability to rapidly adapt to external pressures through diverse tolerance mechanisms, both in vitro and in vivo[61–65]. To assess whether prolonged CPTH2 exposure could induce tolerance that might compromise its synergy with CAS, we subjected *C. auris* strains CBS12767 and yCB799 to 14 days of CPTH2 treatment under both in vitro and in vivo conditions, following

**Fig. 7 | Synergistic antifungal effect of CPTH$_2$ and CAS against *C. auris* in vitro and in vivo. a** Histone H3 acetylation levels in *C. auris* CBS12767 were assessed by Western blot following treatment with increasing concentrations of CPTH$_2$ or DMSO control. Results are representative of three independent experiments. **b** Spot assays show the individual and combined antifungal effects of CPTH$_2$ (0.5 μM) and CAS (31.25 ng/mL for CBS12767; 4 μg/mL for yCB799) against *C. auris* CBS12767 and yCB799. **c, d** Checkerboard assays evaluated CPTH$_2$ and CAS combinations against CBS12767 (**c**) and yCB799 (**d**). FICI values are shown in the upper right corners. **e, g** Time-kill curves for CPTH$_2$ and CAS, alone or in combination, were generated for CBS12767 (**e**) and yCB799 (**g**). Combo1 and Combo2 indicate 4 μM CPTH$_2$ with 7.8 or 31.25 ng/mL CAS (**e**), and 6 μM CPTH$_2$ with 8 or 128 μg/mL CAS (**g**). Data are representative of three independent experiments. **f, h** Colony formation after 48-h drug exposure (alone or combined) and plating on drug-free SDA medium, followed by 36-h incubation at 30 °C for CBS12767 (**f**) and yCB799 (**h**). **i, j** Checkerboard assays demonstrate synergistic activity of CPTH$_2$ and CAS against clinical isolates from five *C. auris* clades (**i**) and the *gcn5Δ*−2 mutant (**j**), with FICI values in the upper right corner; I, II, III, IV, and V represent Clades I–V. **k** Survival analysis of *Galleria mellonella* larvae (*n* = 20) infected with yCB799 and treated with CPTH$_2$ alone (upper panel) or CPTH$_2$ and CAS alone or in combination (lower panel). **l** Fungal burden measured in the spleens and liver of mice infected with yCB799 after 6 treatments with CPTH$_2$ (10 mg/kg, *n* = 6; 40 mg/kg, *n* = 8) and CAS (10 mg/kg), either alone or in combination. Data presented in (**a**–**k**) are expressed as mean ± SD. Statistical significance analysis was performed using Log-rank (Mantel-Cox) test (**k**) or one-way ANOVA with Sidak's test (**a, l**). Source data are provided as a Source Data file.

established protocols[62]. We then evaluated: (1) CPTH$_2$-CAS synergy using checkerboard assays and (2) Inhibitory effects of CPTH$_2$ on Gcn5-mediated histone H3 acetylation via Western blot analysis (Fig. S8a). Checkerboard assays revealed that following 14 days of induction both in in vitro and in vivo, neither CBS12767 nor yCB799 developed resistance to the drug CPTH$_2$ · CAS synergy (Fig. S8b−e, Table S3, 4). Western blot analysis further confirmed that CPTH$_2$ consistently inhibited Gcn5-mediated histone H3 acetylation under these conditions (Fig. S8f−i). Interestingly, in CBS12767, histone H3 acetylation remained susceptible to CPTH$_2$ at concentrations of 4 μM or lower during in vitro experiments. However, extended exposure to 8 μM CPTH$_2$ led to development of resistance to its inhibitory effects (Fig. S8f), with the resistance starting to manifest around day 10 (Fig. S8j). Whether this reflects an evolved resistance within Gcn5 itself or the activation of compensatory pathways remains an open question, warranting further investigation.

## Evaluating the safety of targeting Gcn5 for antifungal therapy against *C. auris* infections

We present evidence suggesting that inhibiting Gcn5 for the treatment of *C. auris* infections is a safe approach. First, our findings demonstrate that *C. auris* Gcn5 exhibits significantly higher susceptibility to the inhibitor CPTH$_2$ compared to host cells. Specifically, CPTH$_2$ effectively inhibits Gcn5 activity in *C. auris* at a concentration of just 1 μM, while a substantially higher concentration of 400 μM is required for inhibition in mammalian cells (Fig. 8a, Fig. S9a, b). This significant disparity indicates a considerable safety margin for utilizing Gcn5 inhibitors in the treatment of *C. auris* infections. Second, at a concentration of 400 μM, CPTH$_2$ only minimally increased lactate dehydrogenase (LDH) release from mammalian cells (Fig. 8b, Fig. S9c). Third, tests involving *Galleria mellonella*, CPTH$_2$ demonstrated no toxicity at a concentration of 1 μg/mL, which is well above its effective therapeutic concentration of 16 ng/mL (Fig. 8c). Finally, CPTH$_2$ was administered intravenously at doses of 10, 20, or 40 mg/kg/day to C57BL/6 mice over four consecutive days, with an equivalent volume of solvent serving as a control. Mice weight was monitored daily, and biochemical markers of liver and kidney function were measured in serum 24 h after the last dose, accompanied by histological examinations of the organs. The results indicated that CPTH$_2$ administration did not lead to weight loss in mice (Fig. 8d), nor did it significantly alter biochemical markers such as ALT, AST, CREA, BUN, and LDH (Fig. 8e), suggesting no acute damage to liver or kidney functions. Furthermore, histological analysis revealed no evidence of damage to the kidneys, liver, spleen, heart, or lungs of the mice (Fig. 8f, Fig. S9d). In summary, these findings indicate that targeting Gcn5 could be a safe strategy for treating *C. auris* infections. However, further research is necessary to confirm these observations.

## Discussion

Addressing the challenge of multidrug-resistant *C. auris* necessitates a thorough understanding of its antifungal resistance mechanisms and pathogenic traits to identify new therapeutic targets. Most research focused on developing novel antifungal agents emphasizes targeting the fungal cell wall or membrane, either directly or indirectly, to disrupt homeostasis, thus inhibiting fungal growth or mitigating drug resistance[35]. This study aims to identify a specific protein that regulates the expression of genes involved in maintaining cell wall and membrane homeostasis in *C. auris* at the transcriptional level. Gene expression in both higher mammals and unicellular fungi is known to be regulated by epigenetics, with PTMs of histone H3 playing a pivotal role. Previous studies have demonstrated a close link between these modifications and drug resistance, as well as pathogenicity in human pathogenic fungi. For instance, the histone H3 acetyltransferase complex SAGA is critical for the pathogenicity and virulence of *C. albicans*, *N. glabrata*, *C. neoformans*, and *A. fumigatus*. Disruption of the gene encoding the acetyltransferase catalytic subunit *GCN5* leads to significant defects in the pathogenicity and drug resistance of *C. albicans* and *N. glabratus*[55,66], as well as reduced pathogenicity in *C. neoformans* and various virulence-related phenotypes in *A. fumigatus*[28,53,67]. Similarly, the knockout of the gene encoding the core module subunit *SPT20* of the SAGA complex markedly diminishes the pathogenicity of *A. fumigatus*[68]. Given the technical challenges associated with manipulating *C. auris* genes, our study focused on the genes responsible for writing histone H3 PTMs to explore their effects on antifungal susceptibility and pathogenicity in *C. auris*, aiming to uncover potential therapeutic targets.

Our study represents the first comprehensive examination of histone H3 PTMs in relation to drug resistance and pathogenicity in *C. auris*. We discovered that Gcn5 plays a crucial role in the resistance of *C. auris* to multiple antifungal agents, including azoles, echinocandins, and polyenes. The role of Gcn5 in azole resistance involves two key mechanisms (Fig. 3). First, it regulates the expression of genes involved in the ergosterol biosynthesis pathway, such as *ERG11*, *ERG1*, *ERG3*, and *ERG25*. Also, it controls *UPC2*, a transcription factor that influences these genes. Disruption of *GCN5* impairs ergosterol synthesis, compromising the integrity of the fungal cell membrane, which explains why the *gcn5Δ* mutant exhibits heightened sensitivity to amphotericin B, a drug that targets ergosterol. Second, Gcn5 is critical for the activation of drug efflux pump genes, including *CDR1*, *SNQ2*, and *MDR1*, which expel azole drugs from the cells, thereby diminishing their efficacy.

Moreover, our study reveals that Gcn5 affects echinocandin resistance in *C. auris* by influencing cell wall integrity and calcium homeostasis, mediated through the transcription factor Cas5 and the calcineurin pathway (Fig. 4). Given the challenges of performing multiple rounds of genetic manipulation in *C. auris*, we opted for an alternative approach by overexpressing *CAS5*, *CNA1* or *CRZ1* in *C. albicans gcn5Δ/Δ* to investigate the interplay between Gcn5, Cas5, and the calcineurin pathway under CAS stress. Our results demonstrated that *CAS5* and *CNA1* overexpression, but not *CRZ1*, significantly enhanced the resistance of the *C. albicans gcn5Δ/Δ* mutant to CAS (Fig. 4d). Interestingly, while Crz1 is the primary transcription factor in

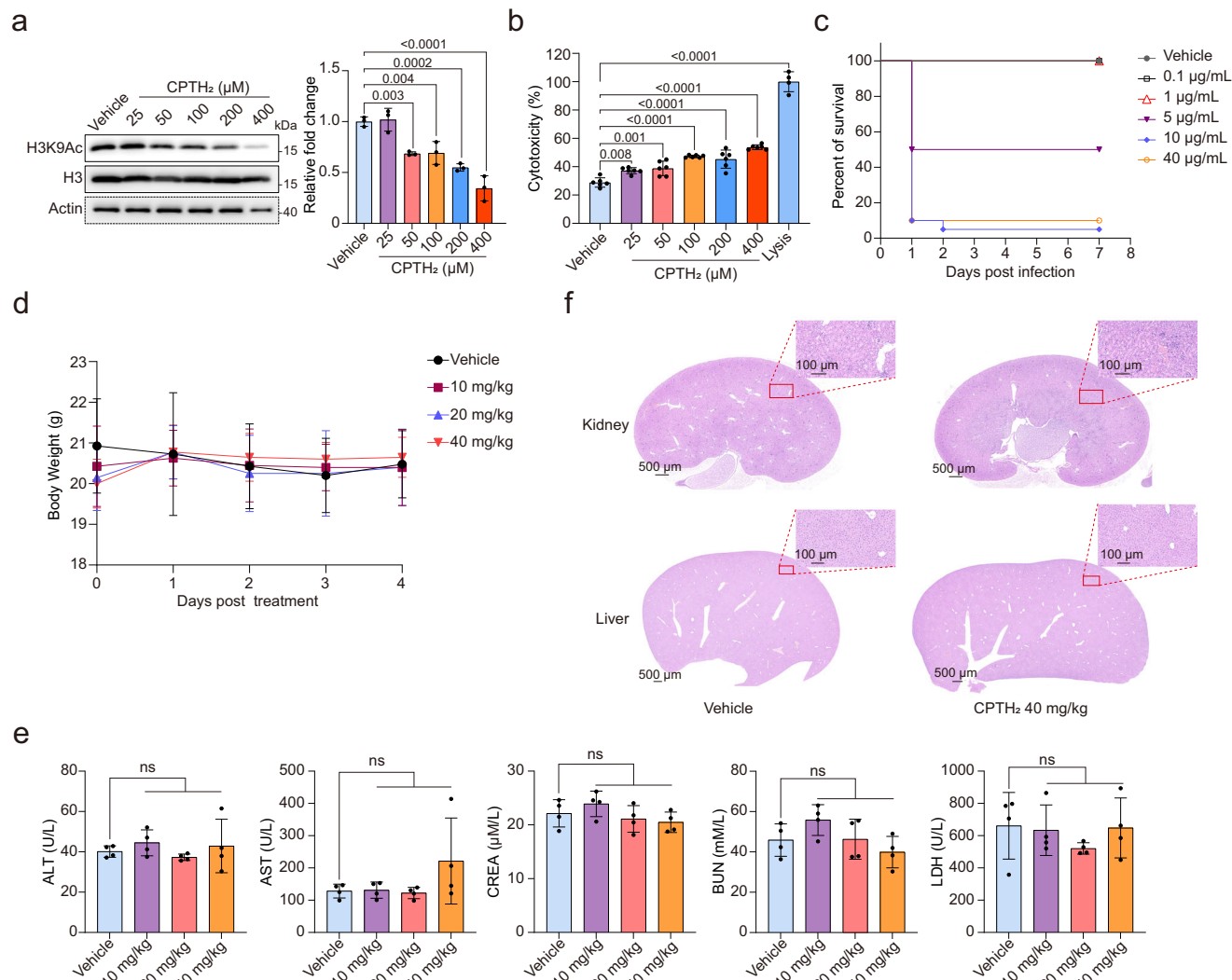

**Fig. 8 | Safety assessment of targeting Gcn5 in the host. a** Western blot analysis of histone H3K9 acetylation levels in HeLa cells treated with various concentrations of CPTH$_2$ or an equivalent volume of DMSO for 24 h. Results are representative of three independent experiments. **b** Detection of LDH release in HeLa cells following treatment with different concentrations of CPTH$_2$ or an equivalent volume of DMSO for 24 h ($n = 6$). Cell lysis solution was added 45 min before detection as a positive control ($n = 4$). Data are presented as the ratio of LDH release in the experimental group to the control group. **c** Survival of *Galleria mellonella* larvae over 7 days following injection with different concentrations of CPTH$_2$ or DMSO ($n = 20$). **d** Body weight monitoring in mice receiving daily intraperitoneal injections of 10, 20, or 40 mg/kg CPTH$_2$, or solvent control, for 4 days ($n = 4$). **e** Serum levels of ALT, AST, CREA, BUN, and LDH measured 24 h after the final injection in the 40 mg/kg group and control mice ($n = 4$). **f** Histological analysis of kidney and liver tissues in all 4 mice from each group, with consistent results observed across individuals. Data presented in (**a–d**) are expressed as mean ± SD. Statistical significance analysis was performed using one-way ANOVA with Sidak's test. Source data are provided as a Source Data file.

the calcineurin pathway, the *C. auris crz1Δ* mutant did not exhibit the heightened CAS sensitivity observed in the *cna1Δ* mutant (Fig. 4c). This suggests that Cna1-mediated echinocandin resistance in *C. auris* is not regulated through Crz1. In contrast, the CAS sensitivity of the *cas5Δ* and *cna1Δ* mutants closely mirrored each other, with both strains also showing comparable hypersensitivity to elevated calcium concentrations (Fig. 4c). Further supporting this connection, overexpression of *CAS5* or *CNA1* in the *C. albicans gcn5Δ/Δ* mutant restored resistance to CAS to a comparable extent (Fig. 4d). Consequently, we hypothesize that the Gcn5-Cna1-Cas5 axis may represent a key epigenetic regulatory pathway responsible for *C. auris* resistance to echinocandins. To confirm this model, further research is warranted to elucidate how Gcn5 regulates Cna1 and whether Cas5 overexpression or activation through phosphorylation site mutations can restore echinocandin resistance in *cna1Δ* mutants. Interestingly, previous studies have indicated that Crz1 does not play a major role in echinocandin resistance in *C. albicans* and *C. tropicalis*[62,63], while it is crucial for *N.*

*glabratus*[69]. Further investigations are required to explore the functional differences of Crz1 across various *Candida* species.

Additionally, our study revealed an intriguing phenomenon: deletion of *GCN5* resulted in opposite changes in MCF resistance between CBS12767 and yCB799. We speculate that this may be a result of at least two contributing factors. First, it is likely that MCF may triggers the expression of uncharacterized MCF-specific, *GCN5*-dependent target genes which are different from that of CAS, and more importantly, these genes function in a strain-dependent manner (Fig. S4c, e). Second, sequencing the *FKS1* and *FKS2* gene loci both in yCB799 and CBS12767 strains identified that only yCB799 *FKS1* gene harbors two mutations (S639F and L906S). S639F is known to be associated with echinocandin resistance[70] and the role of L906S remains unclear. It is possible that *FKS1* mutation may be attributed to the different susceptibility of the two strains to MCF. However, somehow the mutation has no impact on CAS action. Possibly the structure differences between MCF and CAS may reflect their distinct roles in gene regulation.

Furthermore, our findings that *gcn5*Δ mutants exhibit significantly reduced pathogenicity and are more readily cleared from host tissues emphasize the role of Gcn5 as a virulence factor (Figs. 5, 6; Fig. S6). Although the exact mechanisms by which Gcn5 influences pathogenicity remain partially elucidated, its involvement in cell wall integrity and calcium ion regulation may account for the observed decrease in virulence (Fig. 4c, f–h). Previous reports indicate that in *C. albicans* and *N. glabratus*, knockout of Gcn5 or the Spt7 or Spt8 subunits of the SAGA complex results in increased sensitivity to phagocytosis and enhanced killing by macrophages[25,66,71].

The slow progress in antifungal drug development highlights the urgent need for innovative therapeutic approaches. Current antifungal strategies typically target fungal cell membranes, cell walls, and stress response pathways[35]. Encouragingly, our research indicates that Gcn5 influences several critical processes, including cell wall structure, ergosterol biosynthesis, and the calcineurin pathway. Given its significant role in both virulence and resistance in *C. auris*, Gcn5 emerges as a promising target for combating invasive infections. Among the Gcn5 inhibitors evaluated, CPTH$_2$ demonstrates notable efficacy by selectively targeting specific histone H3 acetylation modifications, such as H3K14Ac, H3K18Ac, and H3K36Ac, while exerting minimal impact on others like H3K9Ac, H3K27Ac, and H3K56Ac (Fig. 7a). This selective inhibition may explain why CPTH$_2$ exhibits a synergistic effect only with CAS, but not with FLC, in both in vitro and in vivo studies. To validate these observations, further research is needed, including site-directed mutagenesis to explore the role of specific Gcn5-regulated histone H3 acetylation sites. Additionally, developing more precise and effective Gcn5 inhibitors could potentially reduce *C. auris* resistance and enhance treatment outcomes. Moreover, *C. auris* not only demonstrates inherent multidrug resistance but can also rapidly develop additional resistance when exposed to antifungal treatments. The ineffective eradication of *C. auris* during therapy poses a risk of disease relapse or treatment failure. Currently, echinocandins represent the primary treatment for *C. auris* infections, but resistance during CAS therapy has been documented[19,20]. Our research highlights that both knockout and targeted inhibition of Gcn5 significantly bolster the fungicidal activity of CAS against *C. auris*, irrespective of whether the strain is sensitive or resistant to echinocandins (Fig. 2d, e; Fig. 7e-h). These findings suggest that combining CAS with Gcn5 inhibition could be a promising strategy for treating invasive *C. auris* infections and mitigating the risk of developing resistance. To assess the long-term viability of this approach, we investigated whether *C. auris* evolves compensatory mechanisms under sustained Gcn5 inhibition. While the wild-type strain CBS12767 developed partial resistance to histone H3 acetylation inhibition after prolonged in vitro exposure to CPTH$_2$ (10 days at high concentrations), no such resistance emerged during in vivo exposure over 14 days. These results imply that when administered at a daily in vivo dosage of 40 mg/kg for a duration of 14 days, CPTH$_2$ continues to be efficacious in targeting histone H3 acetylation in *C. auris*. Nevertheless, under extreme in vitro conditions where the concentration of CPTH$_2$ is 8 µM or higher, resistance to the inhibition of histone H3 acetylation can arise within 10 days. This vividly demonstrate the extraordinary adaptability of *C. auris* under intense stress. As a result, it emphasizes the necessity for further investigation into the underlying mechanisms involved, which should be crucial for devising more effective strategies to impede the global dissemination of *C. auris*.

A significant challenge in antifungal drug development is differentiating between human and fungal eukaryotic mechanisms to ensure targeted treatment[35]. Gcn5, an enzyme highly conserved across various species, raises concerns regarding drug specificity. Nevertheless, preliminary safety assessments indicate that Gcn5 in *C. auris* is more susceptible to the inhibitor CPTH$_2$ compared to human cells, with no acute toxicity detected in mice (Fig. 8; Fig. S9). Relevant studies also show that silencing Gcn5 expression or using MB-3 to inhibit Gcn5 activity does not significantly affect the viability or maturation of human cells. However, these approaches effectively enhance the therapeutic response to all-trans retinoic acid in non-acute promyelocytic leukemia acute myeloid leukemia[72]. These findings suggest that targeting Gcn5 could be an effective strategy for treating invasive *C. auris* infections while minimizing host toxicity.

In conclusion, our study elucidates the critical role of histone H3 PTMs in the resistance and pathogenicity of *C. auris*. We provide compelling evidence that targeting Gcn5 significantly potentiates the efficacy of echinocandin treatments, thereby opening new avenues for innovative antifungal strategies against *C. auris* infections.

# Methods

## Ethics statement
All animal experiments were conducted in strict adherence to the Regulations for the Care and Use of Laboratory Animals established by the Ministry of Science and Technology of the People's Republic of China. The study protocol received thorough review and approval from the Institutional Animal Care and Use Committee (IACUC) at the Shanghai Institute of Immunity and Infection, Chinese Academy of Sciences (Permit Number: A2023024).

## Animals
Female ICR mice (6–8 weeks old, weighing 24–26 g) and female C57BL/6 mice (6–8 weeks old, weighing 18–20 g) were procured from Beijing Vital River Laboratory Animal Technology Company (Beijing, China). The mice were housed in a pathogen-free environment maintained at 21 °C with 50–70% relative humidity and a 12 h light/dark cycle. They were provided with unrestricted access to food and water throughout the study. All experimental procedures were conducted in accordance with the protocol approved by the Institutional Animal Care and Use Committee (IACUC) at the Shanghai Institute of Immunity and Infection, Chinese Academy of Sciences, China. The study did not involve sex-based analysis.

## *C. auris* strains, cell lines, and culture medium
Detailed information regarding the strains and plasmids utilized in this study is presented in Supplementary Data 1. The *C. auris* strains used in this study represent multiple clades, including Clade I (CBS12767, yCB799, BJCA001, CBS12766, CBS15605), Clade II (CBS10913, CBS12373, B11220), Clade III (BJCA002, RICU2, B11221), Clade IV (B11244), and Clade V (AR1097). For culturing, *C. auris* strains were grown on Yeast Extract Peptone Dextrose (YPD) medium, with or without the addition of 2% agar. To revive strains from -80 °C storage, they were streaked onto YPD agar plates in quadrants to isolate individual clones and incubated at 30 °C for 48 h. Single colonies were subsequently transferred to a YPD liquid medium and cultured overnight at 30 °C with shaking at 200 rpm until saturation. The following day, cultures were diluted to an initial optical density (OD$_{600}$) of 0.1 in fresh YPD liquid medium and incubated at 30 °C with shaking at 200 rpm for 5 h prior to experimental use.

All cell lines used in this study (HeLa, ATCC CCL-2; Caco-2, ATCC HTB-37; HUVEC, ATCC CRL-1730; J774A.1, ATCC TIB-67) were obtained from the American Type Culture Collection (ATCC) and were cultured in Dulbecco's Modified Eagle's Medium (BMDM) supplemented with 10% fetal bovine serum (FBS) and 1% penicillin-streptomycin. Cells were seeded at densities of $1 \times 10^6$ cells per well in 6-well plates or $5 \times 10^4$ cells per well in 96-well plates and were used for experiments 12 h post-seeding. All cell cultures were incubated at 37 °C in a 5% CO$_2$ atmosphere and maintained under humidified conditions.

## Strains Construction
The method for constructing mutant strains adheres to our previously published protocol[73]. Briefly, approximately 1000 bp regions flanking

the target gene open reading frames (ORFs) were amplified from the genomic DNA of the CBS12767 or yCB799 strains. The NAT1 resistance cassette was amplified from the pSFS2A plasmid. Conventional fusion PCR was employed to combine the upstream and downstream homologous arms with the NAT1 resistance cassette. For constructing the *GCN5* gene complement strain, approximately 1000 bp of homologous arms flanking the *LEU2* gene and the *GCN5* gene expression cassette, including the ORF along with ~1500 bp upstream and 300 bp downstream regions, were amplified using genomic DNA from the CBS12767 strain. The hygromycin B resistance cassette was amplified from the pBARGPE1-Hygro-mCherry plasmid. These components were subsequently inserted into the KpnI/XbaI sites of the pUC19 plasmid using NEBuilder® HiFi DNA Assembly Master Mix. The *GCN5* gene complement components were then PCR-amplified. For constructing the *LEU2* and *GCN5* double knockout strain, we first used the SAT1-Flipper system to construct the *gcn5Δ* strain and excised the SAT1-Flipper cassette under 2% maltose induction. Subsequently, the NAT1 resistance cassette was used to knockout the *SET1* gene. Transformation of *C. auris* was performed using a GenPulser Xcell™ electroporation system (BioRad) following the manufacturer's instructions. The construction of *GCN5* knockout and *CAS5*, *CNA1* or *CRZ1* overexpression strains in *C. albicans* followed our previously published protocol[74]. Briefly, genomic DNA from the SC5314 strain was utilized to amplify ~350 bp of the upstream and 390 bp of the downstream homologous arms flanking the *GCN5* gene ORF. The *LEU2* or *HIS1* gene expression cassettes were amplified from the pSN40 or pSN52 plasmids, respectively. Conventional fusion PCR was then employed to join these homologous arms with the *LEU2* or *HIS1* cassettes. The resulting construct was used to sequentially knock out both copies of the *GCN5* gene in the SN152 parental strain through chemical transformation, with selection on the appropriate deficient media. For constructing *CAS5*, *CNA1* or *CRZ1* gene overexpression strains in the *gcn5Δ/Δ* background, a fragment containing the NAT resistance gene and the TDH3 promoter was amplified from the bSN147 plasmid. Two rounds of PCR were performed to add 88 bp homologous arms to both ends of this fragment. The PCR product was then transformed into the *gcn5Δ/Δ* strain via chemical transformation. Primers used for PCR amplification and validation were synthesized by Genewiz (Jinweizhi Biotechnology Co., Ltd.) and are detailed in Supplementary Data 2.

### Protein extraction and Immunoblotting

Total protein from *C. auris* was extracted using a previously established method[75]. Cells exhibiting an OD$_{600}$ of 1.5 were harvested, washed with ice-cold sterile deionized water, and resuspended in 1 mL of ice-cold deionized water. Subsequently, 150 μL of lysis solution, comprising 138.75 μL of 2 N NaOH and 11.25 μL of β-mercaptoethanol, was added. The mixture was maintained on ice for 30 min, with vortexing every 5 min. Following this incubation, 150 μL of 55% (w/v) trichloroacetic acid (TCA) was introduced, and the mixture was incubated on ice for an additional 30 min, again with vortexing every 5 min. The samples were then centrifuged at 4 °C at maximum speed for 20 min. The supernatant was discarded, and the cell pellet was washed with 50 μL of ice-cold acetone. After the acetone was removed via centrifugation, the pellet was resuspended in 70 μL of HU buffer containing 0.1 M DTT. The samples were heated in a metal bath at 67 °C for 15 min, with vortexing every 5 min, followed by centrifugation at room temperature at maximum speed for 2 min. The supernatant was subsequently collected for immunoblotting.

For host cell protein extraction, the culture medium was discarded, and the cells were washed with 1 mL of ice-cold PBS. Next, 200 μL of 2× Laemmli loading buffer (composed of 65.8 mM Tris-HCl, pH 6.8, 2.1% SDS, 26.3% (w/v) glycerol, 0.01% bromophenol blue, and 5% β-mercaptoethanol) was added. The cells were scraped off using a 1 mL pipette tip and transferred into a clean 1.5 mL tube on ice. The samples were centrifuged at

4 °C at maximum speed for 10 min, and the supernatant was collected. Finally, the samples were heated in a metal bath at 100 °C for 10 min prior to use in immunoblotting.

Western blot images were processed using Photoshop CC (V14.0). Gray value analysis of immunoblot bands was performed using ImageJ (V2.3.0). Uncropped and unprocessed scans of all blots presented in this study are provided in the Source Data file. The types and catalog numbers of the antibodies utilized are provided in Table S5.

### *C. auris* total RNA extraction and qRT-PCR

Following sample collection, 400 μL of ice-cold lysis buffer (composed of 0.1 M LiCl, 0.1 M Tris [pH 7.5], 5% SDS, and 2% β-mercaptoethanol) was added to resuspend the samples. Subsequently, 200 μL of acid-washed glass beads and 200 μL of PCIA (pH 4.5, P120617, Aladdin) were incorporated into the mixture. This combination was vortexed at maximum speed at 4 °C for 5 min, then placed on ice for an additional 5 min. Following this step, the mixture was centrifuged at maximum speed at 4 °C for 10 min, and the supernatant was carefully collected. To purify the supernatant, it was sequentially extracted with 200 μL of PCIA and then with 300 μL of chloroform. After purification, 1 mL of 100% ethanol was added to precipitate the RNA, which was then incubated at -80 °C for 30 min. The mixture was subsequently centrifuged at maximum speed at 4 °C for 10 min, and the resulting pellet was washed once with 75% ethanol. Finally, the pellet was dissolved in 40 μL of nuclease-free water.

cDNA synthesis was performed utilizing a PrimeScript RT Reagent Kit (RR047A, Takara). Quantitative PCR (qPCR) was conducted with TB Green Primer Ex Taq (RR820A, Takara) on an ABI 7900HT Fast Real-Time PCR System. Gene expression levels were normalized to *ADH1*. The primers employed for qRT-PCR were synthesized by Genewiz (Jinweizhi Biotechnology Co., Ltd.) and are detailed in Supplementary Data 2.

### Minimum inhibitory concentration assay

Antifungal susceptibility was assessed using a broth microdilution protocol in flat-bottom 96-well microtiter plates, as outlined in ref. 76, and adhering to the M27-A4 standard method established by the Clinical and Laboratory Standards Institute (CLSI).

### Checkerboard assay

To prepare a checkerboard assay, a 400× stock solution of CPTH$_2$ or CAS was dissolved in a YPD liquid medium to achieve a final concentration of 4×, using an equal volume of DMSO as a control. Subsequently, 50 μL of both CPTH$_2$ and CAS solutions were added to designated wells in a 96-well plate, resulting in a final concentration of 2× for each compound. The overnight culture of test strains was then diluted to ~2×10$^4$ CFU/mL in YPD medium, with 100 μL of this dilution added to each well, bringing the concentration of CPTH$_2$ or CAS to 1×. Following incubation at 30 °C for 36 h, the OD$_{600}$ was measured using a microplate reader, with results normalized against the DMSO control wells. The fractional inhibitory concentration index (FICI) was calculated based on 80% growth inhibition using the formula: FICI = (MIC$_{CPTH2\ Combo}$)/ (MIC$_{CPTH2\ Alone}$) + (MIC$_{CAS\ Combo}$)/(MIC$_{CAS\ Alone}$).

### Time-kill curve

Cells cultured overnight were adjusted to an OD$_{600}$ of 0.05 in YPD liquid medium, corresponding to ~1×10$^6$ CFU/mL. Aliquots of this suspension were transferred into sterile 1.5 mL Eppendorf tubes, to which appropriate concentrations of antifungal drugs or an equivalent volume of DMSO (as a control) were added. Subsequently, 200 μL of the treated cell suspension was dispensed into each well of a 96-well plate, which was then incubated at 30 °C without agitation. Samples were collected from the wells at 0, 2, 4, 8, 24, and 48 h. These samples were serially diluted in tenfold increments and plated onto an SDA

solid medium devoid of antifungal agents. After incubation at 30 °C for 48 h, colonies were counted. Additionally, at the 48 h time point, 100 µL of the original solution was directly plated on an SDA solid medium, incubated at 30 °C for 48 h, and photographed. Each experiment was performed in triplicate.

## Rhodamine 6-G (R-6G) efflux assay

Overnight cultures were adjusted to an $OD_{600}$ of 0.2 in 15 mL of YPD liquid medium and incubated at 30 °C with shaking at 200 rpm for 5 h. Subsequently, cells were harvested by centrifugation, washed three times with sterile PBS, and resuspended in 10 mL of sterile PBS. To induce glucose starvation, cells were incubated at 30 °C with shaking at 200 rpm for an additional 3 h. Following this, the cells were collected, washed again three times with sterile PBS, and resuspended in sterile PBS, adjusting the concentration to $1 \times 10^8$ CFU/mL. A 10 mM stock solution of R-6G was added to achieve a final concentration of 10 µM, and the cells were incubated at 30 °C with shaking at 200 rpm for 2 h to facilitate R-6G uptake. After the incubation period, cells were harvested by centrifugation, washed three times with sterile PBS, and resuspended in 10 mL of sterile PBS. To initiate the efflux process, 500 µL of 40% glucose was added, resulting in a final concentration of 2%. The cells were then incubated at 30 °C with shaking at 200 rpm for 45 min. Following this incubation, the cells were centrifuged to remove the supernatant, and photographs were taken to document any observable changes in cell color. The cells were washed three times with sterile PBS and resuspended in 1 mL of sterile PBS. The red fluorescence intensity was subsequently observed and photographed using the RFP channel of an inverted fluorescence microscope. The images were processed using Photoshop CC (V14.0).

## ChIP-qPCR for *C. auris*

Cells cultured overnight were adjusted to an $OD_{600}$ of 0.1 in 100 mL of YPD liquid medium and incubated at 30 °C with shaking at 200 rpm for 4 h. Following treatment with 256 µg/mL FLC or an equivalent volume of DMSO, cells were cross-linked with formaldehyde to a final concentration of 1% at 30 °C with shaking at 90 rpm for 15 min. The cross-linking reaction was quenched by adding glycine to a final concentration of 0.125 M, followed by incubation at 30 °C with shaking at 90 rpm for an additional 5 min. The cells were then centrifuged at 4 °C at 3724 × g for 5 min, discarding the supernatant. The cell pellet was washed three times with ice-cold PBS while vortexing. Cells were subsequently resuspended in 1 mL of ice-cold PBS and transferred to a 2 mL MP bead-beating tube. Following a centrifugation step at 3724 × g and 4 °C for 2 min, the supernatant was removed. The cells were flash-frozen in liquid nitrogen for 15 min and stored at -80 °C.

The next day, 30 µL of Dynabeads Protein G were transferred to a clean 1.5 mL tube. The supernatant was removed using a magnetic stand (DynaMag™-2, Invitrogen). The beads were washed three times with 800 µL of PBST (PBS with 0.05% Tween-20), rotating at 4 °C for 10 min for each wash. After washing, the beads were resuspended in 400 µL of PBST containing 2 µg of either H3K14Ac (ab52946, Abcam) or H3K9Ac (ab32129, Abcam) antibody or 2 µg of Normal Rabbit IgG (2729S, CST). The mixture was incubated with rotation at 4 °C for 6 h.

The frozen cell samples were thawed on ice and resuspended in 700 µL of Lysis Buffer (50 mM HEPES, 140 mM NaCl, 1 mM EDTA [pH 8.0], 1% Triton X-100, 0.1% deoxycholate, 0.1% SDS, and 1× Protease Inhibitor Cocktail). ZrO2 beads were added, and cells were lysed using a FastPrep-24™ 5 G instrument (MP) with the following settings: 6 m/sec for 60 s, repeated for 4 cycles, with ice bath cooling for 5 min between cycles. After lysis, the mixture was rotated at 4 °C for 30 min. Subsequently, a sterile syringe needle was used to puncture the bottom of the tube, which was then placed into a pre-chilled flow tube and centrifuged at 4 °C at 2095 × g for 5 min. The contents of the flow tube were mixed by pipetting and aliquoted into clean 1.5 mL TPX microtubes (C30010010, Diagenode), with 300 µL allocated per tube. DNA

was sheared using a Diagenode Bioruptor® Plus set to high power mode (30 s ON, 30 s OFF) for 4 cycles. Following centrifugation at 13,800 × g and 4 °C for 15 min, the supernatant was collected into a clean 1.5 mL tube, reserving 60 µL as Input and storing it at -80 °C. Protein concentration was measured using the BCA Protein Assay Kit (T9300A, Takara) with 3 µL of the sample.

For immunoprecipitation, 500 µL of the sample was mixed with 900 µL of RIPA Buffer (200 mM NaCl, 1% Triton X-100, 0.1% deoxycholate, 10 mM Tris-HCl [pH 8.0], 1 mM EDTA [pH 8.0], and 1× Protease Inhibitor Cocktail). Dynabeads were removed from the 4 °C rotator, and the supernatant was discarded. The beads were washed three times with 800 µL TBST by rotation at 4 °C for 10 min. The sample was then added to the tube containing antibody- or Normal Rabbit IgG-bound Dynabeads and incubated by rotation at 4 °C overnight.

The following day, the supernatant was discarded. The beads were sequentially washed with Buffer A (1% Triton X-100, 0.1% SDS, 0.1% deoxycholate, 10 mM Tris-HCl [pH 8.0], 1 mM EDTA [pH 8.0]), Buffer B (1% Triton X-100, 0.1% SDS, 0.1% deoxycholate, 300 mM NaCl, 10 mM Tris-HCl [pH 8.0], 1 mM EDTA [pH 8.0]), Buffer C (250 mM LiCl, 0.5% deoxycholate, 0.5% NP-40, 10 mM Tris-HCl [pH 8.0], 1 mM EDTA [pH 8.0]), Buffer D (0.2% Triton X-100, 10 mM Tris-HCl [pH 8.0], 1 mM EDTA [pH 8.0]), and 1× TE (pH 8.0). Each wash was performed by rotation at 4 °C for 10 min. After the final wash, 100 µL of 1× TE was added to the beads. Input samples were thawed on ice, supplemented with 40 µL of 1× TE to a total volume of 100 µL, and treated with 4 µL of 10% SDS and 4 µL of 5 M NaCl. The mixture was incubated in a shaking metal bath at 65 °C at 600 rpm overnight. The following day, 2 µL of 10 mg/mL RNase was added and incubated at 37 °C at 600 rpm for 1 h. Subsequently, 2.2 µL of 0.5 M EDTA, 4.4 µL of 1 M Tris-HCl (pH 6.5), and 1.1 µL of Proteinase K were added and incubated in a shaking metal bath at 55 °C at 600 rpm for 2 h.

The immunoprecipitated (IP) samples were placed on a magnetic stand, and the supernatant was transferred to a new clean 1.5 mL tube. The beads were washed once more with 100 µL of 1× TE containing 0.5 M NaCl, and the washes were combined. For Input samples, the mixture was centrifuged at 21500 × g and 4 °C for 5 min, and the supernatant was collected and supplemented to 200 µL with 100 µL of 1× TE containing 0.5 M NaCl. The samples were purified using the GeneJET PCR Purification Kit (K0702, Thermo Scientific) and subsequently used for qPCR analysis. The ratio of IP to Input samples was calculated based on CT values. Primers utilized for ChIP-qPCR were synthesized by Genewiz (Jinweizhi Biotechnology Co., Ltd.) and are detailed in Supplementary Data 2.

## Cell Wall Chitin Staining

Overnight cultured cells were diluted to an $OD_{600}$ of 0.1 in 10 mL of YPD liquid medium and incubated at 30 °C with shaking at 200 rpm for 5 h. The culture was then transferred to a 15 mL centrifuge tube, and the cells were collected by centrifugation. Following two washes with PBS, the cell concentration was adjusted to $1 \times 10^7$ CFU/mL using PBS. 1 mL of this suspension was transferred to a sterile 1.5 mL EP tube. The cells were then centrifuged and resuspended in 1 mL of 4% paraformaldehyde for fixation at room temperature for 1 h. After fixation, cells were collected again by centrifugation, washed three times with PBS, and resuspended in 500 µL of 30 µg/mL Calcofluor White (CFW). The suspension was incubated in the dark at 37 °C with shaking for 1 h. Subsequent to incubation, the cells were centrifuged at 1500 × g for 5 min at room temperature, and the supernatant was discarded. The cells underwent three additional washes with PBS, then were resuspended in 100 µL of PBS and pipetted onto a clean microscope slide. The slide was allowed to sit undisturbed for 20 min before the liquid was aspirated. The slide was air-dried away from light. Ten microliters of ProLong Gold anti-fade reagent were then added to the cell smear, which was subsequently covered with a coverslip, gently pressed to

spread the sample, and sealed. Fluorescence was observed and documented using the UW channel of an inverted fluorescence microscope. The images were processed using Photoshop CC (V14.0).

## Cell cytotoxicity assay

Cells were seeded at a density of $5 \times 10^4$ cells per well in a 96-well plate and incubated at 37 °C with 5% $CO_2$ for 12 h. After the culture medium was removed, each well was treated with 100 μL of fresh DMEM medium containing varying concentrations of $CPTH_2$. The cells were then incubated at 37 °C with 5% $CO_2$ for an additional 24 h. Cell cytotoxicity was assessed using the Promega CytoTox 96® Non-Radioactive Cytotoxicity Assay Kit, which quantifies lactate dehydrogenase (LDH) release. Lysis Solution was added to positive control wells to achieve the appropriate working concentration, and the plate was incubated at 37 °C with 5% $CO_2$ for 45 min. Following this, 50 μL of supernatant from each well was transferred to a new 96-well plate, with DMEM medium serving as the negative control. Subsequently, 50 μL of CytoTox 96® Reagent was added to each well, mixed thoroughly, and any bubbles were removed using a clean syringe needle. The plate was then incubated in the dark at room temperature for 10–15 min, after which absorbance was measured at 492 nm using a BioTek Synergy H1 plate reader.0

## RNA-Seq assays

Isolates of *C. auris* CBS12767 and the *gcn5Δ* mutant (three independent clones each) were inoculated into 5 mL of YPD liquid medium and cultured overnight at 30 °C with shaking at 200 rpm until saturation. The cultures were then diluted to an $OD_{600}$ of 0.1 in 10 mL of fresh YPD medium and grown at 30 °C with shaking at 200 rpm for 4 h. Following this, 5 mL of the culture was divided equally into two 15 mL centrifuge tubes. Each tube was treated with either 256 μg/mL fluconazole (FLC) or an equivalent volume of solvent. The cultures were incubated at 30 °C with shaking at 200 rpm for an additional 2 h. After incubation, the cultures were centrifuged, and the cell pellets were collected and quickly frozen in liquid nitrogen. Total RNA was extracted and sent to the Beijing Genomics Institute (BGI) for quality assessment, strand-specific mRNA library preparation, and sequencing. Raw sequencing reads were processed using Trim Galore to remove adapters and low-quality sequences. The cleaned reads were then mapped to the *C. auris* B8441 genome (version s01-m01-r04, Candida Genome Database) using STAR[77]. Gene expression levels were quantified using feature-Counts, and differential expression analysis was performed using DESeq2[78]. Genes with an adjusted *P*-value < 0.05 and |log2 fold change| > 1 were considered differentially expressed. Gene Ontology (GO) enrichment analysis was conducted using clusterProfiler[79].

## Galleria mellonella infection model

The method for constructing the *Galleria mellonella* infection model follows the procedure described in ref. [80]. Briefly, larvae weighing ~250–350 mg, obtained from Tianjin Huiyu Biological Technology Co., Ltd. (Tianjin, China), were used for infection experiments. Overnight cultures of *C. auris* were adjusted to an $OD_{600}$ of 0.1 in 10 mL of YPD liquid medium and incubated at 30 °C with shaking at 200 rpm for 5 h. The cells were then collected by centrifugation at room temperature ($1500 \times g$ for 5 min), washed three times with sterile PBS, and the cell concentration was adjusted to $1 \times 10^8$ CFU/mL. Using a micro syringe, each larva was injected in the center of the left rear proleg with $1 \times 10^6$ cells (10 μL). A total of 20 larvae were used per group. For testing the therapeutic effects of combining $CPTH_2$ and CAS, these drugs were added to the cell suspension immediately before injection. Post-infection, the larvae were placed in disposable culture dishes and maintained in a dark environment at 30 °C, with mortality recorded daily. For fungal burden determination, larvae were placed in tissue homogenization tubes containing 1 mL of sterile PBS 24 or 48 h post-infection. The larvae were homogenized using a tissue homogenizer,

then serially diluted and plated onto SDA solid medium containing ampicillin and gentamicin. The plates were incubated at 30 °C for 48 h before colony counting.

## Mouse systemic infection model

This study utilized 6–8 week-old female ICR mice weighing 24–26 g to assess survival curves and fungal burden in invasive *C. auris* infections, following the protocol described in reference in ref. [56]. Briefly, for survival curve assays, mice received an intraperitoneal injection of 200 mg/kg cyclophosphamide (Sigma-Aldrich, PHR1404-1G) and a subcutaneous injection of 250 mg/kg cortisone acetate (MCE, HY-17461A) 2 days prior to infection and 3 days post-infection, respectively. To evaluate fungal burden in organs, mice were treated with cyclophosphamide and cortisone acetate only 2 days before infection. Overnight cultures of *C. auris* were transferred to 50 mL of YPD liquid medium at an $OD_{600}$ of 0.1 and incubated at 30 °C with shaking at 200 rpm for 5 h. After centrifugation at $1500 \times g$ at room temperature, the cells were washed three times with sterile PBS and resuspended to a concentration of $2.5 \times 10^8$ CFU/mL for survival curve assays or $1 \times 10^8$ CFU/mL for fungal burden assays. Mice were intravenously injected with 200 μL of the fungal suspension. For fungal burden measurements, organs, including kidneys, spleen, liver, brain, and lungs, were collected 3 days post-infection. The organs were weighed, homogenized, and diluted serially. The diluted samples were plated on SDA plates, incubated at 30 °C for 48 h, and colonies were enumerated. For survival curve analysis, mice were regularly weighed and monitored. Mice that became non-viable or unable to access food and water were euthanized, and the time of death was recorded.

For drug efficacy testing, 6–8 week-old female C57BL/6 mice weighing 18–20 g were employed. Mice were immunosuppressed 2 days prior to infection using the same protocol and subsequently infected with $2 \times 10^7$ CFU of *C. auris*. Treatment commenced 4 h post-infection, with subsequent doses administered every 12 h. The number of doses varied by experiment. Fungal burdens in the kidneys, liver, spleen, and lungs were assessed 12 h after the final dose, following the same protocol described above. Caspofungin (MCE, HY-17006), $CPTH_2$ (MCE, HY-W013274), and fluconazole (Sigma-Aldrich, PHR1160-1G) were used in the study. Drugs were prepared as 20× stock solutions in DMSO. For experiments involving only caspofungin and fluconazole, these drugs were diluted to a 1× concentration with sterile PBS. $CPTH_2$ was prepared in a solution containing 10% DMSO, 40% PEG300, 5% Tween-80, and 45% PBS.

## Statistical Analysis

The figure legends detail statistical parameters, including the exact sample size (n) and measures of statistical significance. Statistical analyses were performed using GraphPad PRISM software version 9.4.1 (GraphPad et al., USA). The specific statistical tests applied are also described in the figure legends.

## Reporting summary

Further information on research design is available in the Nature Portfolio Reporting Summary linked to this article.

## Data availability

The authors declare that the data supporting the findings of this study are available within the article and its Supplementary Information files. RNA-Seq data can be found under the GEO accession number GSE293594. Source data are provided with this paper.

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

## Acknowledgements

The authors would like to express their gratitude to Professor Linqi Wang from the Institute of Microbiology, Chinese Academy of Sciences; Professor Guanghua Huang from Huashan Hospital, Fudan University; and Professor Fitz Gerald Salazar Silao from the Wenner-Gren Institute, Science for Life Laboratory, Stockholm University, for providing the strains utilized in this study. We also extend our appreciation to all the lab members at the Shanghai Institute of Immunity and Infection, Chinese Academy of Sciences, and the School of Basic Medical Sciences, Jiangxi Medical College, Nanchang University, for their invaluable assistance during the preparation and discussion of this manuscript. Additionally, we acknowledge the technical support and experimental instruments provided by the technical platform at the Shanghai Institute of Immunity and Infection. This work is financially supported by the MOST Key R&D Program of China (2022YFC2304700, Y.W.), the National Natural Science Foundation of China (32170195 and 32311530119, C.C.; 32060040 and 32460049, X.T.H.; 32300167, Y.W.; 32470200, X.H.H.; 32200161, W.X.); Shanghai Science and Technology Innovation Action Plan 2023 "Basic Research Project" (23JC1404201, C.C.); Shanghai "Belt and Road" Joint Laboratory Project (22490750200, C.C.); the Foundation of State Key Laboratory of Pathogen and Biosecurity (SKLPBS2236, C.C.); the School of Basic Medical Sciences, Nanchang University; the Project for high and talent of Science and Technology Innovation in Jiangxi "Double-Thousand Talents Program of Jiangxi Province" (jxsq2023201019, X.T.H.); the National Key R&D Program of China (2022YFC2303504, X.H.H.); and The Natural Science Foundation of Basic Research Program in Jiangsu Province (BK20231511, W.X.).

## Author contributions

C.C., X.T.H., D.Z., and L.Z. conceived and designed the study. C.C., X.T.H., D.Z., Y.P.Z., L.Z., and M.M. performed data analysis and wrote the manuscript. Y.P.Z., Y.Z., and S.X. conducted all experiments and performed statistical analyses of the data. G.C. carried out an RNA-seq analysis. C.C., X.T.H., D.Z., L.Z., Y.P.Z., X.H.H., Y.W., W.X., Y.D., Y.T., M.G., L.H., and Z.Y. engaged in discussions regarding the experiments and results.

## Competing interests

The authors declare no competing interests.
