## [Transparent Peer Review file · Nature Communications]

Targeting epigenetic regulators to overcome drug resistance in the emerging human fungal pathogen *Candida auris*

Corresponding Author: Dr Changbin Chen

Version 0:

Reviewer comments:

Reviewer #1

(Remarks to the Author)

In this manuscript, Zhang et al. begin by performing a comprehensive analysis of histone H3 posttranslational modifications in the human fungal pathogen *Candida auris*. They do so in response to diverse antifungal drugs as well as upon genetic impairment of the acetyltransferases GCN5 and RTT109, as well as the methyltransferases SET1, SET2, and DOT1. Through these investigations, they identify a key role for Gcn5-mediated H3 acetylation in regulating both azole and echinocandin resistance. Follow-up mechanistic studies suggest the importance of Gcn5 in regulating azole resistance is due to its impact on regulating the expression of ergosterol biosynthesis genes and efflux transporters. For echinocandin resistance, *C. auris* Gcn5 regulates the expression of genes involved in the calcineurin stress response pathway as well as the transcription factor Cas5. Finally, the authors highlight the importance of GCN5 in *C. auris* virulence in both invertebrate and mouse infection models, and show that pharmacological inhibition of Gcn5 potentiates the echinocandin caspofungin *in vitro* and *in vivo*.

Overall, the manuscript is incredibly thorough and well-written. The authors conduct extensive genetic analyses and highlight a novel role for Gcn5 in regulating antifungal resistance in an important human fungal pathogen. They also provide proof-of-principle that targeting this acetyltransferase may serve as a promising antifungal strategy. Many of their key findings are supported by experiments in multiple *C. auris* backgrounds strengthening their conclusions. In order to further increase the impact of their work, I have listed some suggestions in the point-by-point comments below:

Major suggestions:

- 1) While I appreciate that the primary focus of the manuscript is on the role of Gcn5 in regulating antifungal resistance, it would strengthen the conclusions in Figure 1 to have complemented strains for *set1*, *set2*, *dot1*, and *rtt109* strains. Alternatively, an independently-generated deletion mutant can be used to examine the impact of these regulators on H3 posttranslational modifications.
- 2) The differential importance of Gcn5 in regulating micafungin susceptibility in different strain backgrounds is interesting! Is this due to differences in gene expression changes in the different strain backgrounds? Does micafungin, compared with caspofungin, have difference effects on inducing the expression of the transcripts prioritized in Figure 4? Would be an interesting addition to measure transcript changes of genes involved in the calcineurin pathway and CAS5 in these different backgrounds.
- 3) The authors show that treatment of mice with up to 40 mg/kg of the Gcn5 inhibitor CPTH2 results in minimal impact on mouse weight, and does not cause damage to mouse organs. However, the authors only test the inhibitor up to 10 mg/kg in a *C. auris* infection model, with moderate effects in combination with caspofungin reported for fungal CFU from mouse kidney and liver. It would be interesting to repeat with a higher concentration of drug to see if the therapeutic benefit can be improved (either quantifying CFU or mouse survival).

Minor suggestions:

- 1) Abstract, line 35: I think "acetylation" should be changed to "deacetylation", since you are describing the consequence of loss of Gcn5.
- 2) Throughout the manuscript there are examples of where "Gcn5" is written in italics with only the first letter capitalized

(Gcn5). This is incorrect nomenclature. Either describe the gene as "GCN5" or describe the protein as "Gcn5". (Example line 262, 433).

3) Please change "Candida glabrata" to "Nakaseomyces glabratus" to reflect the updated nomenclature classification.

4) When describing the conclusions for Figure 6 and S6, change the conclusion to be specific for the echinocandin drug class, not to "antifungal" drugs (lines 340-341).

5) Add scale bars to microscopy images.

Reviewer #2

(Remarks to the Author)

This study presents a highly significant analysis of the connection between histone H3 modification and antifungal resistance in the recently emerged and highly resistant fungal pathogen *Candida auris*. The story moves from a broad survey of relevant genes and phenotypes to zero in on the acetyltransferase Gcn5. It is this emphasis on Gcn5 that is remarkably comprehensive and significant. The authors show that Gcn5 is required to promote resistance to all major antifungal drug classes, and trace its impact through RNAseq and overexpression-rescue analysis to transcription factor Cas5 and the calcineurin pathway. All this is solid detective work. What sets the study apart from most others is the testing of synergy of *gcn5D* deletion mutations with antifungals in vivo in animal infection models, and the successful use of Gcn5 inhibitor CPTH2 in such animal experiments as well. I am not quite ready to start taking CPTH2 myself, but the authors make a convincing case that this compound or derivatives will be useful for treating *C. auris* infection in a combination therapy modality, and that the susceptibility of *C. auris* cells is much greater than that of mammalian cells or tissues.

Gcn5 has been well studied in fungal pathogens previously (and in just about every other eukaryote). The significance of this particular study as I see it comes from the detailed analysis of Gcn5 in this poorly understood pathogen and the use of the Gcn5 inhibitor, especially in the in vivo experiments.

The presentation is quite good - data are clear; text is intelligible and suitably economical.

1. The one scientific issue that needs to be addressed is the unusual complementation strategy and potential impact on phenotypes in the GCN5 AB strain. There is a vague statement around line 181: "This discrepancy may be attributed to disruptions in the LEU2 gene resulting from the integration of the GCN5 ORF (Fig. 2A)." As far as I can tell the authors complemented defects by disrupting LEU2, so their WT and mutant strains are Leu+ yet the complemented strain is Leu-. This is sloppy. The authors say repeatedly how tough it is to manipulate *C. auris*, but frankly they could have integrated the complementing construct into a non-coding region. Anyway, I recognize that remaking the complemented strain and then re-doing all the experiments would be prohibitive. However, I think that the authors should make a *leu2D* deletion mutant derivative of the *gcn5D* mutant and show that the *leu2* mutation does not affect drug sensitivity. This will be a neat and tidy solution.

2. A minor issue; scientific dotting of i's. ~line 252 the authors measure expression of FKS1 and FKS2 in response to caspofungin treatment. For completeness' sake it would be interesting to see expression of RHO1, the gene that encodes the FKS-associated Ras protein that is required for glucan synthase activity.

3. There is a reference that should be added in my opinion: <https://pubmed.ncbi.nlm.nih.gov/31311209/> In this study Tscherner and Kuchler show that CPTH2 is active against CTG clade *Candida* species. It is not the world's greatest paper, and does not even nick the significance of the manuscript under review, but it is a precedent and one that suggests narrow organismal specificity of the compound.

Reviewer #3

(Remarks to the Author)

This manuscript explores the role of histone-modifying enzymes in the resistance and pathogenicity of *Candida auris*. Growing evidence indicates that epigenetic writers and erasers, responsible for the post-translational modifications (PTMs) of histones, play a crucial role in microbial pathogenesis. In this study, selected candidates were shown to contribute to the antifungal resistance of *C. auris*. The main hypothesis posits that selectively targeting these enzymes could alter the antifungal resistance and pathogenicity of *C. auris*.

To test this, the authors employed a loss-of-function approach to examine the roles of selected acetyltransferases (Gcn5 and Rtt109) and methyltransferases (Set1, Set2, and Dot1) in antifungal drug sensitivity and resistance mechanisms. Mutants deficient in these enzymes exhibited varying degrees of sensitivity to antifungal drugs. Notably, *gcn5* was found to regulate key genes involved in ergosterol biosynthesis and drug efflux, as well as influence cell wall integrity and echinocandin resistance by modulating the calcineurin signaling pathway.

Deletion of *gcn5* significantly reduced the virulence of *C. auris* in infection models, namely *Galleria mellonella* and immunocompromised mice. Moreover, the study demonstrated that the GCN5 inhibitor CPTH2, when combined with the antifungal drug CAS, exhibited a synergistic effect against *C. auris* without significant toxicity to host cells.

Comments

1. The introduction offers a broad overview of antifungal resistance in fungal pathogens but would benefit from a sharper focus on the rationale for targeting epigenetic regulators as a strategy to combat drug resistance in *Candida auris*.

Highlighting the novelty and significance of epigenetic regulation in overcoming antifungal resistance would make the introduction more engaging and provide a clearer justification for the study's objectives.

2. While the inhibition of Gcn5 has shown promise as a strategy to counter infection and antifungal resistance, it is likely that *Candida auris* will evolve compensatory mechanisms to mitigate the loss of this enzyme's activity. Conducting experiments to track the time course of *C. auris* adaptation to Gcn5 inhibition could provide valuable insights into the sustainability and long-term efficacy of this potential therapeutic approach.

3. The loss of Gcn5 function could also impact the activity of non-histone target proteins in *Candida auris*. However, the role of these non-histone proteins was largely overlooked in this study, leaving an important aspect of Gcn5's function unexplored.

4. It remains unclear whether the observed changes in cell wall integrity, drug efflux, and calcineurin signaling are direct consequences of Gcn5 loss or arise indirectly through downstream effects.

5. The calcineurin signaling pathway was investigated for its role in maintaining fungal cell wall integrity and echinocandin resistance. Consistently, deletion of the gene encoding the calcineurin catalytic subunit (CNA1) or the transcription factor CAS5 in both *C. auris* CBS12767 and yCB799 backgrounds resulted in enhanced sensitivity to CAS, a phenotype similar to that of *gcn5* Δ mutants. Additionally, calcineurin signaling genes were downregulated in mutants lacking *gcn5*. Previous studies have also demonstrated that *gcn5*-deficient mutants in *C. glabrata* and *C. neoformans* exhibit heightened sensitivity to the calcineurin inhibitor FK506. Interestingly, overexpression of CAS5 partially restored CAS resistance in the absence of GCN5.

However, the authors have not provided information on the acetyltransferase activities of GCN5 in *C. auris* strains deficient in calcineurin signaling. Without such evidence, it would be premature to conclusively establish the role of the calcineurin pathway in *Gcn5*-dependent echinocandin resistance.

6. Similar to the results observed in mice, was the increased survival of *Galleria mellonella* also correlated with a reduced fungal burden when infected with the *gcn5* mutant or its inhibitors?

7. Line 101: Include a reference

8. Fig.1. Indicate GCN5 AB in the figure legend.

Version 1:

Reviewer comments:

Reviewer #1

(Remarks to the Author)

The authors did an excellent job at addressing all of my original comments, as well as the suggestions from the other reviewers. I recommend the manuscript be accepted to Nature Communications.

Reviewer #2

(Remarks to the Author)

The authors have done an excellent job of addressing my criticisms.

Reviewer #3

(Remarks to the Author)

I have no further comments.

Response to the reviewers

Reviewer #1:

Major suggestions:

Q1. *While I appreciate that the primary focus of the manuscript is on the role of Gcn5 in regulating antifungal resistance, it would strengthen the conclusions in Figure 1 to have complemented strains for set1 Δ , set2 Δ , dot1 Δ , and rtt109 Δ strains. Alternatively, an independently-generated deletion mutant can be used to examine the impact of these regulators on H3 posttranslational modifications.*

Answer: We are truly grateful to the reviewer for the perceptive comment. We agree that without complementation experiments, it would be premature to attribute the loss of histone H3-associated PTMs solely to the knockout of these genes. Following the reviewer's suggestion, we independently re-knock out the genes *SET1*, *SET2*, *DOT1*, and *RTT109* in the *C. auris* CBS12767 background and analyzed histone H3 PTMs via Western blot. We obtained consistent results, as the *set1 Δ* strains (*set1 Δ #1* and *set1 Δ #2*; Fig. 1g) exhibited a nearly complete loss of H3K4 methylation, the *set2 Δ* strains (*set2 Δ #1* and *set2 Δ #2*; Fig. 1h) showed a similar loss of H3K36 methylation, and the *dot1 Δ* strains (*dot1 Δ #1* and *dot1 Δ #2*; Fig. 1i) displayed a nearly complete loss of H3K79 methylation. Additionally, H3K56 acetylation was markedly reduced in the *rtt109 Δ* strains (*rtt109 Δ #1* and *rtt109 Δ #2*; Fig. 1f). These results provide strong evidence for the conserved functions of these genes in the regulation of histone H3 post-translational modifications (PTMs) in *C. auris*. **(Lines 146–148 on page 6; Fig. 1f–i)**

Q2. *The differential importance of Gcn5 in regulating micafungin susceptibility in different strain backgrounds is interesting! Is this due to differences in gene expression changes in the different strain backgrounds? Does micafungin,*

compared with caspofungin, have difference effects on inducing the expression of the transcripts prioritized in Figure 4? Would be an interesting addition to measure transcript changes of genes involved in the calcineurin pathway and CAS5 in these different backgrounds.

Answer: We appreciate the reviewer's concern regarding the unexpected findings in our study (**Lines 187 on page 8**). To clarify whether it is the differential expression of the GCN5-dependent downstream targets in CBS12767 and yCB799 backgrounds that caused different effects of Gcn5 on micafungin susceptibility, we followed the reviewer's suggestion by treating CBS12767, *gcn5Δ*, GCN5 AB, yCB799, and *gcn5Δ-2* strains with caspofungin (CAS) or micafungin (MCF), and performing qRT-PCR to examine the transcriptional levels of genes affected by GCN5 deletion, including *FKS1* and *FKS2* (direct targets of CAS and MCF), the transcription factors CAS5 and CRZ1, the phosphatase *GLC7* (known to facilitate Cas5 nuclear localization), and several Cas5-regulated downstream genes (*ALS1*, *CHS2*, *BMT3*, *PGA52*, *RBE1*, and *UTR2*). Surprisingly, we found that after treatment with either CAS or MCF, the expression patterns of these genes were NOT influenced by the strain background (**Fig. 4a, b and e; Fig. S4a-e**), suggesting that the observed different effects of Gcn5 on micafungin susceptibility may not be due to the differential expression of the GCN5-dependent downstream targets in CBS12767 and yCB799 backgrounds. In other words, the different strain backgrounds appear to have no effect on the phenotype.

However, we found that the expression of these genes was differently depending on whether CAS or MCF was used for treatment. In particular, Cas5-regulated genes, such as *CHS2*, *ALS1*, *PGA42*, and *RBE1*, were significantly upregulated upon CAS treatment but remained largely unchanged under MCF treatment (**Lines 302-309 on page 12, Fig. S4c, e**). Interestingly, their expression patterns were independent of strain background. As per the

reviewer's comment, our data vividly demonstrate that in comparison to CAS, MCF has differential impacts on the induction of the expression of the *GCN5*-regulated downstream target genes, especially for those regulated by Cas5 (but the expression patterns of these genes behave the same in both strains). Certainly, the differences in gene expression patterns between CAS and MCF were consistent across both the CBS12767 and the yCB799 strain backgrounds. This suggests that despite both drugs targeting β -1,3-glucan synthase, they induce distinct transcriptional responses in *C. auris*. It is likely that MCF may triggers the expression of uncharacterized *GCN5*-dependent target genes which are different from that of CAS, and more importantly, these genes function in a strain-dependent manner. Future work will be focusing on the functional analysis of those strain-dependent, MCF-specific and *GCN5*-regulated genes. **(Lines 502-506 on page 19)**

Another possibility for the distinct MCF resistance profiles might be the genomic differences between CBS12767 and yCB799 strains. Previous studies have shown that in *N. glabratus*, *FKS1* or *FKS2* deletion alone does not affect viability, but point mutations in their hot spot (HS) regions confer decreased susceptibility to echinocandins¹⁻⁴. Similarly, *FKS1* and *FKS2* mutations also contribute to resistance of *C. auris*, indicating functional redundancy akin to *N. glabratus*^{5,6}. We therefore sequenced the *FKS1* and *FKS2* gene sets in both CBS12767 and yCB799 strains, and identified that only yCB799 *FKS1* gene harbors two mutations (S639F and L906S) **(Lines 309-312 on page 12, Fig. S4f)**. S639F is known to be associated with echinocandin resistance and the role of L906S remains unclear. It is possible that *FKS1* mutation may be attributed to the different susceptibility of the two strains to MCF. However, somehow the mutation has no impact on CAS action. Possibly the structure differences between MCF and CAS may reflect their distinct role in gene regulation. **(Lines 506-512 on page 19)**

In summary, at least two possibilities may explain the opposing effects of *GCN5* deletion on MCF resistance in CBS12767 and yCB799: (1) MCF may triggers the expression of uncharacterized *GCN5*-dependent target genes which are different from that of CAS, and more importantly, these genes function in a strain-dependent manner; (2) Compared to CBS12767, yCB799 genome harbors two *FKS1* mutations, which may contribute to the different susceptibility of the two strains to MCF. These factors collectively help explain the observed differences in MCF resistance between the two strain backgrounds.

References

- 1 Hou, X. *et al.* Novel FKS1 and FKS2 modifications in a high-level echinocandin resistant clinical isolate of *Candida glabrata*. *Emerging microbes & infections* **8**, 1619-1625 (2019).
- 2 Niimi, K. *et al.* Reconstitution of high-level micafungin resistance detected in a clinical isolate of *Candida glabrata* identifies functional homozygosity in glucan synthase gene expression. *Journal of antimicrobial chemotherapy* **67**, 1666-1676 (2012).
- 3 Pham, C. D. *et al.* Role of FKS mutations in *Candida glabrata*: MIC values, echinocandin resistance, and multidrug resistance. *Antimicrobial agents and chemotherapy* **58**, 4690-4696 (2014).
- 4 Katiyar, S. K. *et al.* Fks1 and Fks2 are functionally redundant but differentially regulated in *Candida glabrata*: implications for echinocandin resistance. *Antimicrobial agents and chemotherapy* **56**, 6304-6309 (2012).
- 5 Lara-Aguilar, V. *et al.* Adaptation of the emerging pathogenic yeast *Candida auris* to high caspofungin concentrations correlates with cell wall changes. *Virulence* **12**, 1400-1417 (2021).
- 6 Ahmed, S. H., El-Kholy, I. M., El-Mehalawy, A. A., Mahmoud, E. M. & Elkady, N. A. Molecular characterization of some multidrug resistant *Candida Auris* in egypt. *Scientific Reports* **15**, 4917 (2025).

Q3. *The authors show that treatment of mice with up to 40 mg/kg of the Gcn5 inhibitor CPTH2 results in minimal impact on mouse weight, and does not cause damage to mouse organs. However, the authors only test the inhibitor up to 10 mg/kg in a C. auris infection model, with moderate effects in combination with caspofungin reported for fungal CFU from mouse kidney and liver. It would be interesting to repeat with a higher concentration of drug to see if the therapeutic benefit can be improved (either quantifying CFU or mouse survival).*

Answer: We sincerely appreciate the reviewer's valuable suggestion and have conducted additional experiments to further evaluate the therapeutic efficacy of CPTH₂ (40 mg/kg) in combination with CAS (10 mg/kg), using a systemic mouse infection model. These experiments included assessments of organ fungal burden and survival analysis.

For the survival study, mice were infected via tail vein injection with the echinocandin-resistant *C. auris* strain yCB799 (5×10^7 CFU per mouse). Immunosuppression was administered two days before and three days after infection. Treatment began 4 hours post-infection, with drugs administered every 12 hours for a total of six doses, followed by survival monitoring. Organ fungal burden was assessed following the same methodology as in **Fig. 7I**.

Evidently, we observed that increasing the dose of CPTH₂ from 10 mg/kg to 40 mg/kg failed to significantly improve therapeutic efficacy, whether CPTH₂ was used as monotherapy or in combination with CAS (**Line 404 on page 15; Fig. 7I, S7d**). Similarly, the results of the survival analysis revealed that the 40 mg/kg CPTH₂ + 10 mg/kg CAS combination did not lead to any further improvement in the survival rate or the length of survival in mice, as compared to CAS monotherapy (**Lines 409–410 on page 15; Fig. S7e**). We therefore postulate that CPTH₂ dose at 10 mg/kg attains the maximal inhibition of *C. auris* Gcn5 activity *in vivo*, such that any further increase in the dose offers no additional advantages.

Minor suggestions

Q1. *Abstract, line 35: I think “acetylation” should be changed to “deacetylation”, since you are describing the consequence of loss of Gcn5.*

Answer: We greatly appreciate the reviewer’s suggestion. We agree that "acetylation" was used incorrectly in this context. Deletion of GCN5, which encodes an acetyltransferase, leads to the loss of histone H3 acetylation, but NOT the effect of enzymatic activity of a deacetylase. We believe that "loss of acetylation" is a more precise description than "deacetylation" (Lines 33–34 on page 2).

Q2. *Throughout the manuscript there are examples of where “Gcn5” is written in italics with only the first letter capitalized (Gcn5). This is incorrect nomenclature. Either describe the gene as “GCN5” or describe the protein as “Gcn5”. (Example line 262, 433).*

Answer: We greatly appreciate the reviewer’s suggestion and apologize for the formatting mistakes regarding gene and protein names. We have carefully gone through the entire manuscript and made all the necessary corrections.

Q3. Please change “Candida glabrata” to “Nakaseomyces glabratus” to reflect the updated nomenclature classification.

Answer: We sincerely appreciate the reviewer’s valuable suggestion. We have revised all instances of “*Candida glabrata*” to “*Nakaseomyces glabratus*” throughout the manuscript.

Q4. When describing the conclusions for Figure 6 and S6, change the conclusion to be specific for the echinocandin drug class, not to “antifungal” drugs (lines 340-341).

Answer: We sincerely appreciate the reviewer's valuable comments. According to the reviewer's suggestion, we have changed "antifungal drug" to caspofungin (CAS) in the conclusions for Fig. 6 and Fig. S6. **(Line 367 on page 14).**

Q5. Add scale bars to microscopy images.

Answer: Corrected in **Fig. 3h** and **Fig. 4g**.

Reviewer #2:

Major comments

Q1. *The one scientific issue that needs to be addressed is the unusual complementation strategy and potential impact on phenotypes in the GCN5 AB strain. There is a vague statement around line 181: "This discrepancy may be attributed to disruptions in the LEU2 gene resulting from the integration of the GCN5 ORF (Fig. 2A)." As far as I can tell the authors complemented defects by disrupting LEU2, so their WT and mutant strains are Leu+ yet the complemented strain is Leu-. This is sloppy. The authors say repeatedly how tough it is to manipulate C. auris, but frankly they could have integrated the complementing construct into a non-coding region. Anyway, I recognize that remaking the complemented strain and then re-doing all the experiments would be prohibitive. However, I think that the authors should make a leu2D deletion mutant derivative of the gcn5D mutant and show that the leu2 mutation does not affect drug sensitivity. This will be a neat and tidy solution.*

Answer: We agree with the reviewer's comment and made changes accordingly. Indeed, it may have problems by disrupting the *LEU2* gene to complement *GCN5*, since the loss of *LEU2* may affect the phenotype. Instead, this problem could be avoided if *GCN5* is reintroduced into a non-coding region. Our initial complementation strategy was based on the approach previously used in *C. albicans* which inserts the wild type copy of candidate gene in the *LEU2* locus of knockout strain, a strain that both copies of candidate gene were replaced with *C.m. LEU2* and *C.d. HIS1*. However, we generated the *gcn5Δ* mutant of *C. auris* by replacing the *GCN5* gene ORF with a NAT selection marker, instead of the *LEU2* marker. Thus, complementing the *GCN5* gene in *LEU2* locus will inactivate the function of gene *LEU2*. To eliminate the potential influence of *LEU2* disruption on antifungal drug sensitivity in *C. auris*, we followed the reviewer's advice and conducted the experiments described as

below.

First, we constructed a *LEU2* knockout strain (*leu2Δ*) in the wild-type CBS12767 background and assessed its drug sensitivity using a spot assay. Our results showed that, as expected, both the *leu2Δ* and *GCN5 AB* strains were unable to grow on leucine-deficient medium, confirming the successful construction of the strains (**Fig. S1c**). Second, we found that the *leu2Δ* strain did not exhibit greater drug-resistant compared the wild-type strain, suggesting that the restored antifungal drug resistance observed after *GCN5* complementation is unlikely due to *LEU2* disruption. (**Lines 176–178 on page 7, Fig. S1c**).

Next, we examined the drug sensitivity of *leu2Δ* in the presence of the Gcn5-specific inhibitor CPTH₂. At 0.5 μM CPTH₂, a concentration that effectively inhibits Gcn5-mediated histone H3 acetylation (**Fig. S1d; Fig. 7a**), we distinctly found that the deletion of *LEU2*, in tandem with the inhibition of Gcn5 activity by CPTH₂, led to a drug sensitivity phenotype that was comparable to the one seen when Gcn5 was inhibited solely by CPTH₂. This finding firmly illustrates that the deletion of *LEU2* is unable to reverse the increased sensitivity to antifungal drug that arises from the loss of *GCN5*. (**Lines 178–182 on page 7, Fig. S1e**).

Finally, we created a double knockout strain (*leu2Δgcn5Δ*) by deleting both the *LEU2* and *GCN5* genes. Once more, we noticed that deleting the *LEU2* gene had no impact on the drug sensitivity profiles of the *gcn5Δ* mutant. This further validates that the restored antifungal drug resistance seen in the *GCN5 AB* strain is unlikely to be due to the disruption of the *LEU2* gene. (**Lines 182-185 on page 7, Fig. S1f**).

Q2. A minor issue; scientific dotting of i's. ~line 252 the authors measure expression of *FKS1* and *FKS2* in response to caspofungin treatment. For

completeness' sake it would be interesting to see expression of RHO1, the gene that encodes the FKS-associated Ras protein that is required for glucan synthase activity.

Answer: We sincerely appreciate the reviewer's valuable comment. As a regulatory subunit of β -1,3-D-glucan synthase, Rho1 is crucial for the activity of Fks1/2. To investigate the impact of Gcn5 on *RHO1* expression, we performed RT-qPCR analysis. Interestingly, similar to *FKS1*, we found that *RHO1* expression was significantly downregulated upon *GCN5* deletion in both CBS12767 and yCB799 backgrounds, regardless of CAS treatment (**Lines 253-258 on page 10, Fig. 4a, S3a**). This finding further reinforces the role of Gcn5 in regulating *C. auris* echinocandin resistance.

Q3. *There is a reference that should be added in my opinion: <https://pubmed.ncbi.nlm.nih.gov/31311209/> In this study Tschermer and Kuchler show that CPTH2 is active against CTG clade Candida species. It is not the world's greatest paper, and does not even nick the significance of the manuscript under review, but it is a precedent and one that suggests narrow organismal specificity of the compound.*

Answer: We sincerely appreciate the reviewer's valuable suggestion. In response, we have added the requested reference to our manuscript (**Line 376 on page 14**).

Reviewer #3:

Comments

*Q1. The introduction offers a broad overview of antifungal resistance in fungal pathogens but would benefit from a sharper focus on the rationale for targeting epigenetic regulators as a strategy to combat drug resistance in *Candida auris*. Highlighting the novelty and significance of epigenetic regulation in overcoming antifungal resistance would make the introduction more engaging and provide a clearer justification for the study's objectives.*

Answer: We sincerely appreciate the reviewer's valuable comment. In response, we have revised the introduction to provide a more concise discussion on the current development of novel antifungal drugs. This revision allows us to more clearly emphasize the rationale for targeting epigenetic regulators as a promising strategy to overcome *C. auris* antifungal resistance and to highlight its potential therapeutic applications **(Lines 75-80, 101-103 on pages 4-5)**.

*Q2. While the inhibition of Gcn5 has shown promise as a strategy to counter infection and antifungal resistance, it is likely that *Candida auris* will evolve compensatory mechanisms to mitigate the loss of this enzyme's activity. Conducting experiments to track the time course of *C. auris* adaptation to Gcn5 inhibition could provide valuable insights into the sustainability and long-term efficacy of this potential therapeutic approach.*

Answer: We sincerely appreciate the reviewer's constructive comment. A defining characteristic of *C. auris* is its ability to rapidly evolve adaptive mechanisms under stress. Therefore, investigating whether *C. auris* develops compensatory mechanisms under prolonged Gcn5 inhibition is crucial for refining our study.

To address this, we carried out both *in vitro* and *in vivo* assays following established protocols (**Lines 414-420 on page 16, Fig. S8a**). For the *in vitro* assay, *C. auris* wild-type strains CBS12767 or yCB799 were cultured at 30°C in YPD medium supplemented with or without 1–8 µM CPTH₂. The cultures underwent serial passages every 48 hours over a period of 14 days. Subsequently, single colonies were isolated and analyzed using checkerboard assays and Western blotting. Each strain and condition were replicated three times independently. For the *in vivo* assay, immunosuppressed mice (cyclophosphamide 200 mg/kg + hydrocortisone acetate 250 mg/kg) were intravenously injected with cells from CBS12767 (2×10^7 CFU/mouse) or yCB799 (1×10^7 CFU/mouse). The mice received daily intraperitoneal injections of CPTH₂ (40 mg/kg) for 14 days. After this treatment period, kidney homogenates were plated, and single colonies were analyzed via checkerboard assays and Western blotting.

The results of our checkerboard assays indicated that following 14 days of induction both in *in vitro* and *in vivo*, neither CBS12767 nor yCB799 developed resistance to the drug CPTH₂ - CAS synergy (**Lines 420-422 on page 16, Fig. S8b-e, Table S3, 4**). Western blot analysis provided additional confirmation that under these circumstances, CPTH₂ persistently suppressed Gcn5-mediated histone H3 acetylation (**Lines 422-424 on page 16, Fig. S8f-i**). Notably, in CBS12767, histone H3 acetylation remained susceptible to CPTH₂ at concentrations of 4 µM or lower during *in vitro* experiments. However, extended exposure to 8 µM CPTH₂ led to development of resistance to its inhibitory effects (**Fig. S8f**), with the resistance starting to manifest around day 10 (**Lines 424-429 on page 16, Fig. S8j**).

These results imply that when administered at a daily *in vivo* dosage of 40 mg/kg for a duration of 14 days, CPTH₂ continues to be efficacious in targeting histone H3 acetylation in *C. auris*. Nevertheless, under extreme *in vitro*

conditions where the concentration of CPTH₂ is 8 μM or higher, resistance to the inhibition of histone H3 acetylation can arise within 10 days. This vividly demonstrate the extraordinary adaptability of *C. auris* under intense stress. As a result, it emphasizes the necessity for further investigation into the underlying mechanisms involved, which should be crucial for devising more effective strategies to impede the global dissemination of *C. auris*. (Lines 541-552 on page 20)

Q3. *The loss of Gcn5 function could also impact the activity of non-histone target proteins in Candida auris. However, the role of these non-histone proteins was largely overlooked in this study, leaving an important aspect of Gcn5's function unexplored.*

Answer: We are truly grateful for the reviewer's perceptive comment. It is well-established that Gcn5 acetylates a plethora of non-histone proteins in addition to histone H3 across diverse fungal species. This includes the budding yeast *Saccharomyces cerevisiae* and phytopathogenic fungi¹⁻⁵. For instance, in the rice blast fungal pathogen *Magnaporthe oryzae*, Gcn5 plays a role in regulating autophagy through the acetylation of Atg7². Moreover, it impacts stress response, energy metabolism, cell toxicity, and cell death via the modification of other proteins³. In *S. cerevisiae*, Gcn5 was found to regulate the autophagic degradation of Rph1, which is crucial for cellular homeostasis, through the acetylation of Rph1⁵. These findings indicate that Gcn5 likely has functions that extend well beyond histone H3 acetylation.

To date, no research has explored whether Gcn5 regulates fungal drug resistance through the acetylation of non-histone proteins. We are extremely grateful for the reviewer's question. Actually, we have noticed this possible function of Gcn5 when we were working on this project. An acetyl-proteomics analysis was carried out to compare the protein acetylation profiles of the wild-type CBS12767 and *gcn5Δ* mutant strains after they had been subjected to a

one-hour treatment with caspofungin (CAS) or DMSO (serving as the control). Our acetyl-proteomics analysis yielded a compelling observation: aside from regulating the transcription of azole- and echinocandin-resistance genes through histone H3 acetylation, Gcn5 might also directly acetylate crucial resistance-related proteins. In comparison to the wild-type strain, the *gcn5Δ* mutant showed substantially decreased acetylation levels in a list of crucial proteins, including the transcription factor Tac1 regulating ABC transporter drug efflux pumps, the azole target Erg11, multiple enzymes (Erg25, Erg6, Erg7) involved in the ergosterol biosynthesis pathway, the echinocandin target Fks1, the transcription factor Cas5, the upstream kinase Cmk1 in the calcineurin pathway, and components of the MAPK signaling pathway (MAPKKK Ssk2 and MAPKK Pbs2) (**Fig a-e**). Our ongoing studies are to elucidate the precise role of these acetylation modifications in Gcn5-mediated *C. auris* drug resistance.

Figure a Heatmap of quantitative analysis of lysine site-specific acetylation modifications across treatment groups.

Figure b Statistical comparison of differentially expressed proteins (quantitative proteomics) and differentially modified lysine acetylation sites (acetylome) across pairwise groups.

Figure c GO enrichment analysis of proteins featuring differentially modified lysine acetylation sites in *gcn5Δ* vs. WT after the treatment with 100 ng/mL CAS

for 1 hour.

Figure d KEGG pathway enrichment analysis of proteins featuring differentially modified lysine acetylation sites in *gcn5Δ* vs. WT after the treatment of 100 ng/mL CAS for 1 hour.

Figure e A list of proteins with differentially modified lysine-acetylated sites associated with the drug resistance of *C. auris*, in *gcn5Δ* vs. WT under the treatment of 100 ng/mL CAS for 1 hour.

References:

- 1 Rössl, A., Denoncourt, A., Lin, M.-S. & Downey, M. A synthetic non-histone substrate to study substrate targeting by the Gcn5 HAT and sirtuin HDACs. *The Journal of Biological Chemistry* **294**, 6227-6239, doi:10.1074/jbc.RA118.006051 (2019).
- 2 Zhang, S. *et al.* Phototrophy and starvation-based induction of autophagy upon removal of Gcn5-catalyzed acetylation of Atg7 in *Magnaporthe oryzae*. *Autophagy* **13**, 1318-1330, doi:10.1080/15548627.2017.1327103 (2017).
- 3 Liang, M. *et al.* Label-Free Quantitative Proteomics of Lysine Acetylome Identifies Substrates of Gcn5 in *Magnaporthe oryzae* Autophagy and Epigenetic Regulation. *MSystems* **3**, doi:10.1128/mSystems.00270-18 (2018).
- 4 Downey, M. *et al.* Gcn5 and sirtuins regulate acetylation of the ribosomal protein transcription factor Ifh1. *Current Biology : CB* **23**, 1638-1648, doi:10.1016/j.cub.2013.06.050 (2013).
- 5 Li, F. *et al.* Gcn5-mediated Rph1 acetylation regulates its autophagic degradation under DNA damage stress. *Nucleic Acids Research* **45**, 5183-5197, doi:10.1093/nar/gkx129 (2017).

Q4. *It remains unclear whether the observed changes in cell wall integrity, drug efflux, and calcineurin signaling are direct consequences of Gcn5 loss or arise*

indirectly through downstream effects.

Answer: We greatly appreciate the reviewer's question. To ascertain whether Gcn5 had a direct regulatory role in cell wall integrity, drug efflux, and calcineurin signaling, we carried out ChIP-qPCR validation experiments. As shown in **Fig. 3e and f**, there is a notable enrichment of H3K14Ac at the promoter regions of the drug efflux pump gene *CDR1* and the azole drug target gene *ERG11*. Remarkably, the enrichment levels of H3K14Ac at these promoter regions significantly increase just after 30 minutes exposure of fluconazole. Moreover, results in **Fig. 1e** provide conclusive evidence that in *C. auris*, nearly all of the H3K14Ac modification is reliant on Gcn5. These findings strongly support that Gcn5 exerts a direct regulatory influence on the transcription of *CDR1* and *ERG11* by means of histone H3 acetylation.

Interestingly, the results from our protein acetylation proteomics analysis revealed that Gcn5 might directly acetylate the transcription factor Tac1, which plays a crucial role in regulating the expression of *CDR1*, and also Erg11. This discovery hints the potential existence of a dual regulatory mechanism, by which Gcn5 could potentially fine-tune drug efflux and the ergosterol biosynthesis. It does so not only through the modification of histone H3 via acetylation but also by directly acetylating essential regulatory proteins.

In order to evaluate the function of Gcn5 in the calcineurin signaling and cell wall integrity, we investigated the enrichment of H3K14Ac at the promoter regions of the TF-encoding genes *CRZ1* and *CAS5* involved in the calcineurin pathway, as well as gene *FKS1*, which encodes the β -1,3-glucan synthase. This examination was carried out under the treatment of CAS at time points of 0, 30, 60, and 120 minutes. Just like genes *CDR1* and *ERG11*, these genes exhibited a notable enrichment of H3K14Ac, peaking at 60 minutes post-treatment (**Lines 321-327 on page 12, Fig. S4g-i**). Moreover, the data from protein acetylation proteomics analysis indicate that Gcn5 may also directly acetylate proteins

Fks1 and Cas5, as well as the upstream kinase Cmk1 of calcineurin Cna1. This additional evidence further reinforces the contribution of a dual regulatory mechanism employed by Gcn5, analogous to its role in drug efflux and ergosterol biosynthesis.

Q5. *The calcineurin signaling pathway was investigated for its role in maintaining fungal cell wall integrity and echinocandin resistance. Consistently, deletion of the gene encoding the calcineurin catalytic subunit (CNA1) or the transcription factor CAS5 in both C. auris CBS12767 and yCB799 backgrounds resulted in enhanced sensitivity to CAS, a phenotype similar to that of gcn5Δ mutants. Additionally, calcineurin signaling genes were downregulated in mutants lacking gcn5. Previous studies have also demonstrated that gcn5-deficient mutants in C. glabrata and C. neoformans exhibit heightened sensitivity to the calcineurin inhibitor FK506. Interestingly, overexpression of CAS5 partially restored CAS resistance in the absence of GCN5.*

However, the authors have not provided information on the acetyltransferase activities of GCN5 in C. auris strains deficient in calcineurin signaling. Without such evidence, it would be premature to conclusively establish the role of the calcineurin pathway in Gcn5-dependent echinocandin resistance.

Answer: We are truly grateful for the reviewer's insightful question and fully agree with the perspective presented. Our existing research findings have indeed established a regulatory connection between Gcn5 and the calcineurin pathway. In our studies on *C. albicans*, we found that overexpression of CAS5, other than CRZ1, both of which are genes encoding transcription factors associated with the calcineurin pathway, was able to partially restore the drug resistance of *gcn5ΔΔ* (**Fig.4d**). Nevertheless, the question of whether Gcn5 regulates the calcineurin pathway or if the calcineurin pathway has an impact on Gcn5 activity remains unresolved.

To tackle this uncertainty, in accordance with the reviewer's suggestion, we initially examined the acetyltransferase activity of Gcn5 in the *cna1Δ* strain. Intriguingly, regardless of whether CAS was administered or not, the acetylation levels of histone H3 at multiple sites remained unchanged in the *cna1Δ* strain compared to those in the wild-type, suggesting that the disruption of the calcineurin pathway had no any effects on Gcn5 activity (287-291 on page 11, Fig. S3d). We therefore speculate that Gcn5 is more likely to function upstream of the calcineurin pathway.

To further substantiate this conclusion, we delved into whether the overexpression of *CNA1* could compensate for the loss of *GCN5*, similar to *CAS5*. Tremendous efforts were made to create the *C. auris* strain in which *CNA1* is overexpressed in the genetic background of *gcn5Δ* strain, but all these attempts proved unsuccessful. Subsequently, we decided to test our hypothesis using the *C. albicans* system instead. In the *gcn5Δ/Δ* mutant strain of *C. albicans*, *CNA1* overexpression is able to partially rescue the growth defect induced by CAS treatment, mirroring the effect observed when *CAS5* was overexpressed in a similar context (Lines 284-287 on page 11, Lines 484-487, 491-494 on page 18, Fig. 4d). This compelling evidence strongly supports our hypothesis that the calcineurin pathway mediates Gcn5-dependent echinocandin resistance.

Q6. *Similar to the results observed in mice, was the increased survival of Galleria mellonella also correlated with a reduced fungal burden when infected with the gcn5 mutant or its inhibitors?*

Answer: We sincerely appreciate the reviewer's insightful suggestions. Although the majority of studies using the *Galleria mellonella* infection model to evaluate fungal pathogenicity mainly depend on survival curves, a few also take into account the assessment of fungal burden. In order to conduct a more thorough and comprehensive analysis, we infected *Galleria mellonella* with

either *C. auris* CBS12767 or *gcn5Δ* at a dose of 7.5×10^5 CFU per larva. Then, we measured the fungal burden at 24 and 48 hours post-infection (**Lines 336-341 on page 13, Fig. S5b**). The obtained results were consistent with the survival curve data and the findings from the mouse model. Specifically, it was evident that the larvae infected with the *gcn5Δ* mutant strain had a significantly lower fungal burden when compared to those infected with the wild-type strain.

Moreover, we infected *Galleria mellonella* with *C. auris* yCB799 at a higher dose of 1×10^6 CFU per larva and administered treatments simultaneously, as illustrated in **Fig. 7k**. When we measured the fungal burden at 24 and 48 hours post-infection, we found that by 48 hours, the treated groups (receiving CPTH₂, CAS, or a combination of both) exhibited a significant lower fungal burdens than the untreated control group. Notably, the combination treatment resulted in an even greater reduction in the fungal burden compared to using CAS alone, which further validates the enhanced antifungal efficacy of the combination therapy (**Lines 398-403 on page 15, Fig. S7c**).

Q7. Line 101: Include a reference

Answer: We thank the reviewer for the input. In response, we have incorporated the relevant references in the corresponding sections of our work. (**Line 91 on page 4**)

Q8. Fig.1. Indicate GCN5 AB in the figure legend.

Answer: We are deeply thankful to the reviewer's comments. We have made the necessary revisions in the figure legend of Fig. 1.